# Synaptic learning rules for sequence learning

Eric Torsten Reifenstein[1,2]*, Ikhwan Bin Khalid[1], Richard Kempter[1,2,3]

[1]Institute for Theoretical Biology, Department of Biology, Humboldt-Universität zu Berlin, Berlin, Germany; [2]Bernstein Center for Computational Neuroscience Berlin, Berlin, Germany; [3]Einstein Center for Neurosciences Berlin, Berlin, Germany

**Abstract** Remembering the temporal order of a sequence of events is a task easily performed by humans in everyday life, but the underlying neuronal mechanisms are unclear. This problem is particularly intriguing as human behavior often proceeds on a time scale of seconds, which is in stark contrast to the much faster millisecond time-scale of neuronal processing in our brains. One long-held hypothesis in sequence learning suggests that a particular temporal fine-structure of neuronal activity — termed 'phase precession' — enables the compression of slow behavioral sequences down to the fast time scale of the induction of synaptic plasticity. Using mathematical analysis and computer simulations, we find that — for short enough synaptic learning windows — phase precession can improve temporal-order learning tremendously and that the asymmetric part of the synaptic learning window is essential for temporal-order learning. To test these predictions, we suggest experiments that selectively alter phase precession or the learning window and evaluate memory of temporal order.

## Introduction

It is a pivotal quality for animals to be able to store and recall the order of events ('temporal-order learning', *Kahana, 1996*; *Fortin et al., 2002*; *Lehn et al., 2009*; *Bellmund et al., 2020*) but there is only little work on the neural mechanisms generating asymmetric memory associations across behavioral time intervals (*Drew and Abbott, 2006*). Putative mechanisms need to bridge the gap between the faster time scale of the induction of synaptic plasticity (typically milliseconds) and the slower time scale of behavioral events (seconds or slower). The slower time scale of behavioral events is mirrored, for example, in the time course of firing rates of hippocampal place cells (*O'Keefe and Dostrovsky, 1971*), which signal when an animal visits certain locations ('place fields') in the environment. The faster time scale is given by the temporal properties of the induction of synaptic plasticity (*Markram et al., 1997*; *Bi and Poo, 1998*) — and spike-timing-dependent plasticity (STDP) is a common form of synaptic plasticity that depends on the millisecond timing and temporal order of presynaptic and postsynaptic spiking. For STDP, the so-called 'learning window' describes the temporal intervals at which presynaptic and postsynaptic activity induce synaptic plasticity. Such precisely timed neural activity can be generated by phase precession, which is the successive across-cycle shift of spike phases from late to early with respect to a background oscillation (*Figure 1*). As an animal explores an environment, phase precession can be observed in the activity of hippocampal place cells with respect to the theta oscillation (*O'Keefe and Recce, 1993*; *Buzsáki, 2002*; *Qasim et al., 2021*). Phase precession is highly significant in single trials (*Schmidt et al., 2009*; *Reifenstein et al., 2012*) and occurs even in first traversals of a place field in a novel environment (*Cheng and Frank, 2008*). Interestingly, phase precession allows for a temporal compression of a sequence of behavioral events from the time scale of seconds down to milliseconds (*Figure 1*; *Skaggs et al., 1996*; *Tsodyks et al., 1996*; *Cheng and Frank, 2008*), which matches the widths of generic STDP learning windows (*Abbott and Nelson, 2000*; *Bi and Poo, 2001*;

*For correspondence:
eric@bccn-berlin.de

Competing interests: The authors declare that no competing interests exist.

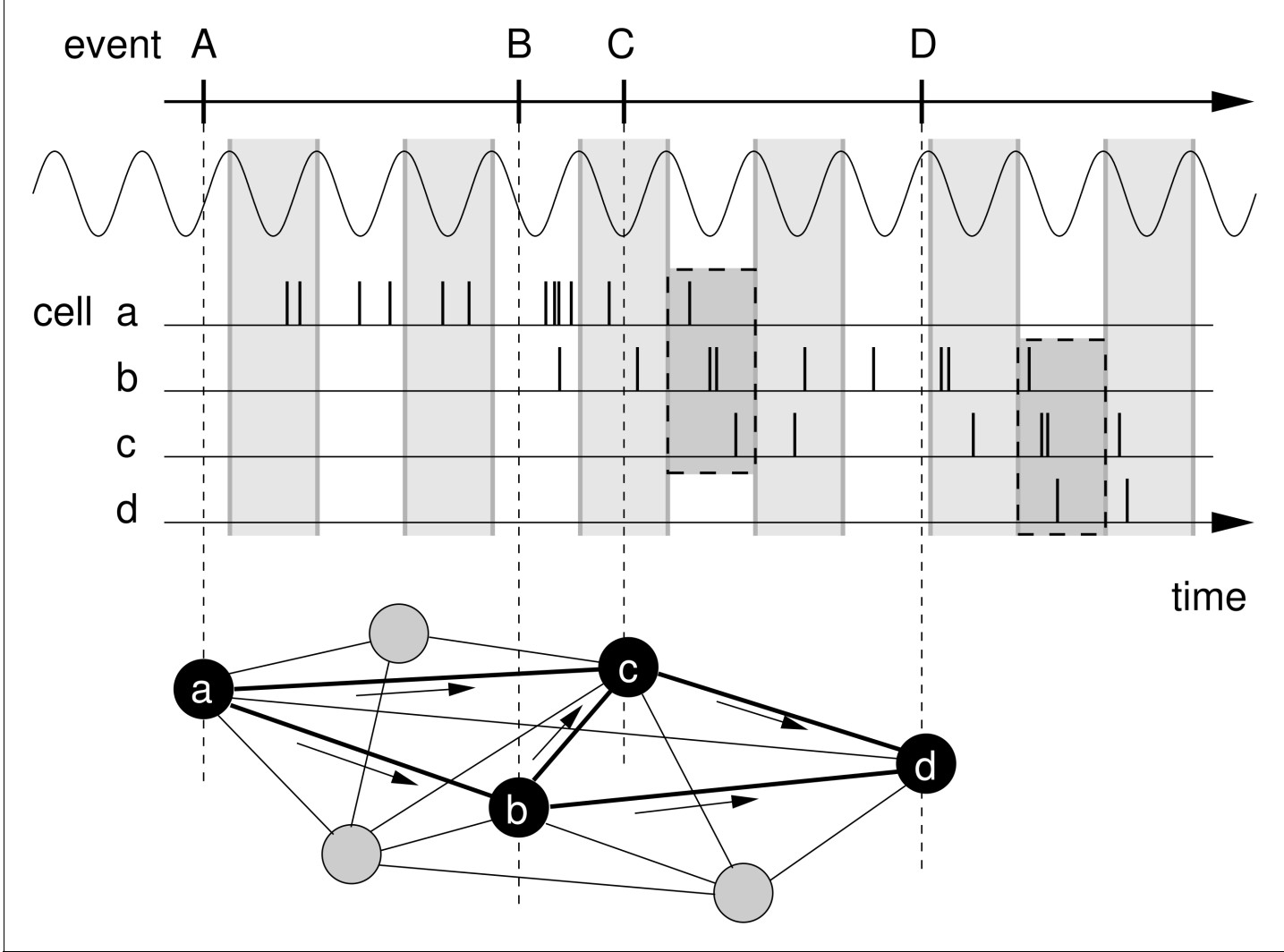

**Figure 1.** Rationale for temporal-order learning via phase precession. Top: Behavioral events (A to D) happen on a time scale of seconds. Middle: These events are represented by different cells (a–d), which fire a burst of stochastic action potentials in response to the onset of their respective event. We assume that each cell shows phase precession with respect to the LFP's theta oscillation (every second cycle is marked by a greay box). When the activities of multiple cells overlap, the sequence of behavioral events is compressed in time to within one theta cycle (two examples highlighted in the dashed, shaded boxes). Bottom: This faster time scale can be picked up by STDP and strengthen the connections between the cells of the sequence. Figure adapted from *Korte and Schmitz, 2016*.

*Froemke et al., 2005*; *Wittenberg and Wang, 2006*). This putative advantage of phase precession for temporal-order learning, however, has not yet been quantified. To assess the benefit of phase precession for temporal-order learning, we determine the synaptic weight change between pairs of cells whose activity represents two events of a sequence. Using both analytical methods and numerical simulations, we find that phase precession can dramatically facilitate temporal-order learning by increasing the synaptic weight change and the signal-to-noise ratio by up to an order of magnitude. We thus provide a mechanistic description of associative chaining models (*Lewandowsky and Murdock, 1989*) and extend these models to explain how to store serial order.

## Results

To address the question of how behavioral sequences could be encoded in the brain, we study the change of synapses between neurons that represent events in a sequence. We assume that the temporal order of two events is encoded in the asymmetry of the efficacies of synapses that connect

neurons representing the two events (**Figure 1**). After the successful encoding of a sequence, a neuron that was activated earlier in the sequence has a strengthened connection to a neuron that was activated later in the sequence, whereas the connection in the reverse direction may be unchanged or is even weakened. As a result, when the first event is encountered and/or the first neuron is activated, the neuron representing the second event is activated. Consequently, the behavioral sequence could be replayed (as illustrated by simulations for example in *Tsodyks et al., 1996*; *Sato and Yamaguchi, 2003*; *Leibold and Kempter, 2006*; *Shen et al., 2007*; *Cheng, 2013*; *Chenkov et al., 2017*; *Malerba and Bazhenov, 2019*; *Gillett et al., 2020*) and the memory of the temporal order of events is recalled (*Diba and Buzsáki, 2007*; *Schuck and Niv, 2019*). We note, however, that in what follows we do not simulate such a replay of sequences, which would depend also on a vast number of parameters that define the network; instead, we rather focus on the underlying change in connectivity, which is the very basis of replay, and draw connections to 'replay' in the Discussion.

Let us now illustrate key features of the encoding of the temporal order of sequences. To do so, we consider the weight change induced by the activity of two sequentially activated cells $i$ and $j$ that represent two behavioral events (dashed lines in **Figure 2A**). Classical Hebbian learning (**Hebb, 1949**), where weight changes $\Delta w_{ij}$ depend on the product of the firing rates $f_i$ and $f_j$, is not suited for temporal-order learning because the weight change is independent of the order of cells:

$$\Delta w_{ij} \propto f_i \cdot f_j = f_j \cdot f_i \propto \Delta w_{ji}.$$

Therefore, a classical Hebbian weight change is symmetric, that is, $\Delta w_{ij} - \Delta w_{ji} = 0$. This result can be generalized to learning rules that are based on the product of two arbitrary functions of the firing rates. We note that, although not suited for temporal-order learning, Hebbian rules are able to achieve more general 'sequence learning', where an association between sequence elements is created — independent of the order of events. To become sensitive to temporal order, we use spike-timing dependent plasticity (STDP; *Markram et al., 1997*; *Bi and Poo, 1998*). For STDP, average weight changes depend on the cross-correlation function of the firing rates (example in **Figure 2C, D**),

$$C_{ij}(t) := \int_{-\infty}^{\infty} dt' \, f_i(t') f_j(t' + t),$$

which is anti-symmetric: $C_{ij}(t) = C_{ji}(-t)$. Assuming additive STDP, that is, weight changes resulting from pairs of pre- and postsynaptic action potentials are added, the average synaptic weight change $\Delta w_{ij}$ between the two cells in a sequence can then be calculated explicitly (*Kempter et al., 1999*):

$$\Delta w_{ij} = \int_{-\infty}^{+\infty} dt \, W(t) C_{ij}(t) \tag{1}$$

where $W$ is the STDP learning window (example in **Figure 2E**). We aim solve **Equation 1** for given firing rates $f_i$ and $f_j$. To do so, we assume that the synaptic weight $w_{ij}$ is generally small and thus only has a weak impact on the cross-correlation of the cells during encoding, that is, for the 'encoding' of a sequence the cross-correlation function is dominated by feedforward input, whereas the recurrent inputs are neglected.

Next, let us show that the symmetry of $W$ is essential for temporal-order learning. Any learning window $W$ can be split up into an even part $W^{\mathrm{even}}$, with $W^{\mathrm{even}}(t) = W^{\mathrm{even}}(-t)$, and an odd part $W^{\mathrm{odd}}$, with $W^{\mathrm{odd}}(t) = -W^{\mathrm{odd}}(-t)$, such that $W = W^{\mathrm{even}} + W^{\mathrm{odd}}$. For even learning windows, one can derive from **Equation 1** and the anti-symmetry of $C_{ij}$ that weight changes are symmetric, that is, $\Delta w_{ij} = \Delta w_{ji}$; therefore, only the odd part $W^{\mathrm{odd}}$ of $W$ is useful for learning temporal order.

To further explore requirements for encoding the temporal order of a sequence of events, we restrict our analysis to odd learning windows. We then can relate the weight change $\Delta w_{ij}$ to the essential features of $C_{ij}(t)$. To do so, we integrate **Equation 1** by parts (with $W$ replaced by $W^{\mathrm{odd}}$),

$$\Delta w_{ij} = \underbrace{\left[ \overline{W^{\mathrm{odd}}}(t) \cdot C_{ij}(t) \right]_{-\infty}^{+\infty}}_{=0} + \int_{-\infty}^{+\infty} dt \left[ -\overline{W^{\mathrm{odd}}}(t) \right] C'_{ij}(t), \tag{2}$$

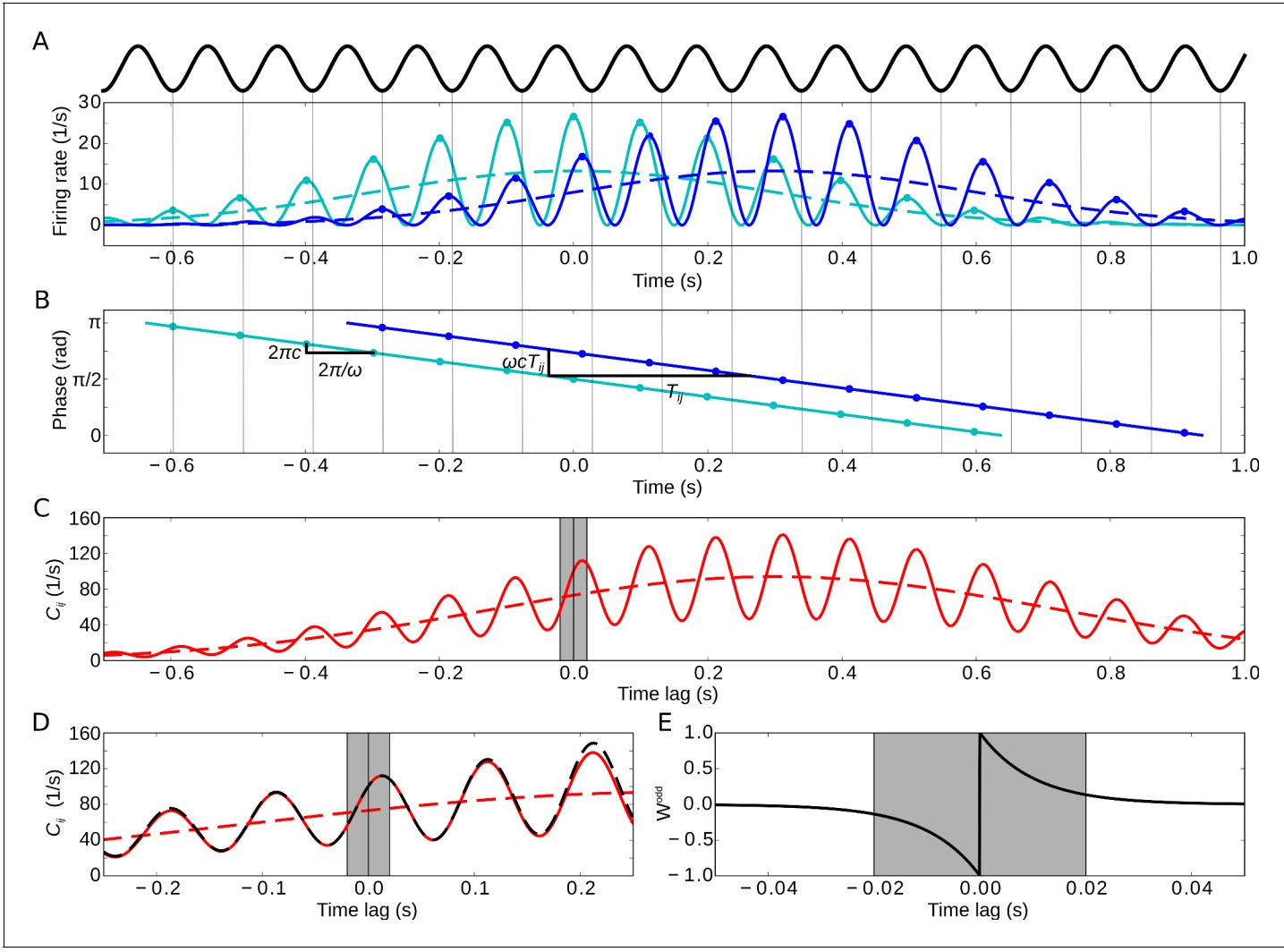

**Figure 2.** Model of two sequentially activated phase-precessing cells. (**A**) Oscillatory firing-rate profiles for two cells (solid blue and cyan lines). The black curve depicts the population theta oscillation. For easier comparison of the two different frequencies, the population activity's troughs are continued by thin gray lines, and the peaks of the cell-intrinsic theta oscillation are marked by dots. Dashed lines depict the underlying Gaussian firing fields without theta modulation. (**B**) Phase precession of the two cells (same colors as in A). The compression factor $c$ describes the phase shift per theta cycle for an individual cell ($2\pi c$). For the temporal separation $T_{ij}$ of the firing fields and the theta frequency $\omega$, the phase difference between the cells is $\omega c T_{ij}$. The dots depict the times of the maxima in (**A**). (**C**) Resulting cross-correlation for the two firing rates from (**A**). The solid red curve shows the full cross-correlation. The dashed line depicts the cross-correlation without theta-modulation. The gray region indicates small ($t<20$ ms) time lags. (**D**) Same as in (**C**), but zoomed in. Note that the first peak of the theta modulation is at a positive non-zero time lag, reflecting phase precession. The dashed black curve shows the approximation of the cross-correlation for the analytical treatment (Materials and methods, **Equation 17**). (**E**) Synaptic learning window. The gray region indicates the region in which the learning window is large, and this region is also indicated in (**C**) and (**D**). Positive time lags correspond to postsynaptic activity following presynaptic activity. Parameters for all plots: $T_{ij} = 0.3$ s, $\omega = 2\,\pi \cdot 10$ Hz, $\sigma = 0.3$ s, $\tau = 10$ ms, $\mu = 1$, $c = 0.042$, $A = 10$.

with the primitive $\overline{W^{\mathrm{odd}}}(t) := \int_{-\infty}^{t} \mathrm{d}t'\, W^{\mathrm{odd}}(t')$ and the derivative $C'_{ij}(t) := \frac{\mathrm{d}}{\mathrm{d}t} C_{ij}(t)$. Because $\overline{W^{\mathrm{odd}}}(t)$ can be assumed to have finite support (note that $\int_{-\infty}^{+\infty} \mathrm{d}t\, W^{\mathrm{odd}}(t) = 0$), the first term in **Equation 2** vanishes. Also the learning window has finite support, and therefore we can restrict the integral in the second term in **Equation 2** to a finite region of width $K$ around zero:

$$\Delta w_{ij} = \int_{|t|<K} \mathrm{d}t \left[ -\overline{W^{\mathrm{odd}}}(t) \right] C'_{ij}(t) \tag{3}$$

where $K$ describes the width of the learning window $W$ (gray region in *Figure 2E*). The integral in *Equation 3* can be interpreted as the cross-correlation's slope around zero, weighted by the symmetric function $-\overline{W^{\mathrm{odd}}}(t)$; interestingly, features of $C_{ij}$ for $|t| \gg K$, for example whether side lobes of the correlation function are decreasing or not, are irrelevant.

As a generic example of sequence learning, let us consider the activities of two cells $i$ and $j$ that encode two behavioral events, for example the traversal of two place fields of two hippocampal place cells. In general, the cells' responses to these events are called 'firing fields'. We model these firing fields as two Gaussian functions $G_{0,\sigma}$ and $G_{T_{ij},\sigma}$ that have the same width $\sigma$ but different mean values 0 and $T_{ij}$ (we note that $T_{ij}$ and $\sigma$ are measured in units of time, that is, seconds; *Figure 2A*, dashed curves). In this case of identical Gaussian shapes of the two firing fields, the cross-correlation $C_{ij}(t)$ is also a Gaussian function, denoted by $G_{T_{ij},\sqrt{2}\sigma}$, but with mean $T_{ij}$ and width $\sqrt{2}\sigma$ (dashed curve in *Figure 2C*). The value $\sigma = 0.3$ s, which we use in the example of *Figure 2*, matches experimental findings on place cells (*O'Keefe and Recce, 1993*; *Geisler et al., 2010*).

It is widely assumed that phase precession facilitates temporal-order learning (*Skaggs et al., 1996*; *Dragoi and Buzsáki, 2006*; *Schmidt et al., 2009*), but it has never been quantitatively shown. To test this hypothesis and to calculate how much phase precession contributes to temporal-order learning, we consider Gaussian firing fields that exhibit oscillatory modulations with theta frequency $\omega$ (*Figure 2A*, solid curves). The time-dependent firing rate of cell $i$ is described by $f_i(t) \propto G_{\mu_i,\sigma}(t)\{1 + \cos[\omega(t - c\mu_i)]\}$, that is, a Gaussian that is multiplied by a sinusoidal oscillation; see also *Equation 11* in Materials and methods. Phase precession occurs with respect to the population theta, which oscillates at a frequency of $(1 - c)\omega$ that is slightly smaller than $\omega$, with a 'compression factor' $c$ that is usually small: $0 \leq c \ll 1$ (*Dragoi and Buzsáki, 2006*; *Geisler et al., 2010*). This compression factor $c$ describes the average advance of the firing phase — from theta cycle to theta cycle — in units of the fraction of a theta cycle; $c$ thus determines the slope $\omega c$ of phase precession (*Figure 2B*). A typical value is $c \approx \pi/(4\sigma\omega)$, which accounts for 'slope-size matching' of phase precession (*Geisler et al., 2010*); that is, $c$ is inversely proportional to the field size $L := 4\sigma$ of the firing field, and the total range of phase precession within the firing field is constant and equals $\pi \equiv 180°$. If there are multiple theta oscillation cycles within a firing field ($\omega\sigma \gg 1$), which is typical for place cells, the cross-correlation $C_{ij}(t)$ is a theta modulated Gaussian (solid curve in *Figure 2C*; see also *Equation 15* in Materials and methods).

The generic shape of the cross-correlation $C_{ij}$ in *Figure 2C* allows for an advanced interpretation of *Equation 3*, which critically depends on the width $K$ of the learning window $W$. We distinguish here two limiting cases: narrow learning windows ($K \ll 1/\omega \ll \sigma$), that is, the width $K$ of the learning window is much smaller than a theta cycle and the width of a firing field, and wide learning windows ($K \gg \sigma$), that is, the width $K$ of the learning window exceeds the width of a firing field. Let us first consider narrow learning windows. Only later in this manuscript, we will turn to the case of wide learning windows.

## Dependence of temporal-order learning on the overlap of firing fields for narrow learning windows ($K \ll 1/\omega \ll \sigma$)

We first show formally that sequence learning with narrow learning windows requires that the two firing fields do overlap, that is, their separation $T_{ij}$ should be less than or at least similar to the width $\sigma$ of the firing fields. In *Equation 3*, which was derived for odd learning windows, the weight change $\Delta w_{ij}$ is determined by $C'_{ij}(t)$ around $t = 0$ in a region of width $K$. For narrow learning windows ($K \ll 1/\omega$), this region is small compared to a theta oscillation cycle and much smaller than the width $\sigma$ of a firing field. Because the envelope of the cross-correlation $C_{ij}(t)$ is a Gaussian with mean $T_{ij}$ and width $\sqrt{2}\sigma$, the slope $C'_{ij}(t = 0)$ scales with the Gaussian factor $G_{T_{ij},\sqrt{2}\sigma}(0) \propto \exp[-T_{ij}^2/(4\sigma^2)]$. The weight change $\Delta w_{ij}$ therefore strongly depends on the separation $T_{ij}$ of the firing fields. When the two firing fields do not overlap ($T_{ij} \gg \sigma$), the factor $\exp[-T_{ij}^2/(4\sigma^2)]$ quickly tends to zero, and sequence learning is not possible. On the other hand, when the two firing fields do have considerable overlap ($T_{ij} \lesssim \sigma$) we have $\exp[-T_{ij}^2/(4\sigma^2)] \lesssim 1$. In this case, sequence learning may be feasible with narrow learning windows. In this section, we will proceed with the mathematical analysis for overlapping fields, which allows us to assume $\exp[-T_{ij}^2/(4\sigma^2)] \approx 1$.

For overlapping firing fields ($T_{ij} \lesssim \sigma$), let us now consider the fine structure of the cross-correlation $C_{ij}(t)$ for $|t| < K$, as illustrated in *Figure 2D*. Importantly, phase precession causes the first positive peak (i.e. for $t > 0$) of $C_{ij}$ to occur at time $c\,T_{ij}$ with $c \ll 1$ (*Dragoi and Buzsáki, 2006*; *Geisler et al., 2010*); phase precession also increases the slope $C'_{ij}(t)$ around $t = 0$, which could be beneficial for temporal-order learning according to *Equation 3*. To quantify this effect, we calculated the cross-correlation's slope at $t = 0$ (see also *Equation 18* in Materials and methods):

$$C'_{ij}(0) \propto G_{T_{ij},\sqrt{2}\sigma}(0) \left[ \frac{T_{ij}}{\sigma} + \omega\sigma\sin(\omega c T_{ij}) + \frac{T_{ij}}{2\sigma}\cos(\omega c T_{ij}) \right]. \tag{4}$$

How does $C'_{ij}(0)$ depend on the temporal separation $T_{ij}$ of the firing fields? If the two fields overlap entirely ($T_{ij} = 0$) the sequence has no defined temporal order, and thus $C'_{ij}(0)$ is zero. For at least partly overlapping firing fields ($T_{ij} \lesssim \sigma$) and typical phase precession where $c = \pi/(4\omega\sigma) \ll 1$, we will show in the next paragraph (and explain in Materials and methods in the text below *Equation 18*) that the second addend in *Equation 4* dominates the other two. In this case, $C'_{ij}(0)$ is much higher as compared to the cross-correlation slope in the absence of phase precession ($c = 0$), leading to a clearly larger synaptic weight change for phase precession. The maximum of $C'_{ij}(0)$ is mainly determined by this second addend (multiplied by $G_{T_{ij},\sqrt{2}\sigma}(0)$) and it can be shown (see Materials and methods) that this maximum is located near $T_{ij} \approx \sqrt{2}\sigma$.

The increase of $C'_{ij}(0)$ induced by phase precession can be exploited by learning windows $W$ that are narrower than a theta cycle (e.g. gray regions in *Figure 2C,D,E*). To quantify this effect, let us consider a simple but generic shape of a learning window, for example, the odd STDP window $W(t) = \mu\,\mathrm{sign}(t)\exp(-|t|/\tau)$ with time constant $\tau$ and learning rate $\mu > 0$ (*Figure 2E*); this STDP window is narrow for $\tau \ll 1/\omega$. *Equations 3 and 4* then lead to (see Materials and methods, *Equation 19*) the average weight change

$$\Delta w_{ij} = A^2 \mu\,\tau^2 \frac{G_{T_{ij},\sqrt{2}\sigma}(0)}{\sigma} \left[ \frac{T_{ij}}{\sigma} + \frac{\omega\sigma\sin(\omega c T_{ij})}{1+\omega^2\tau^2} + \frac{T_{ij}}{2\sigma}\cos(\omega c T_{ij}) \cdot \frac{1-\omega^2\tau^2}{(1+\omega^2\tau^2)^2} \right] \tag{5}$$

where $A$ depicts the number of spikes per field traversal. Note that, according to *Equation 3*, the weight change $\Delta w_{ij}$ in *Equation 5* can be interpreted as a time-averaged version of $C'_{ij}(t)$ near $t = 0$ from *Equation 4*. Thus, *Equations 4 and 5* have a similar structure, but *Equation 5* includes multiple incidences of the term $\omega^2\tau^2$ that account for this averaging. This term is small for narrow learning windows ($\tau \ll 1/\omega$) and can thus be neglected ($\omega^2\tau^2 \ll 1$) in this limiting case; however, for typical biological values of $\tau \geq 10$ ms and $\omega = 2\pi \cdot 10$ Hz, the peculiar structure of the $\omega^2\tau^2$-containing factor in the third addend in the square brackets is the reason why this addend can be neglected compared to the first one; as a result, the cases of 'phase locking' ($c = 0$) and 'no theta' (only the first addend remains) are basically indistinguishable. Moreover, for narrow odd learning windows, $\Delta w_{ij}$ in *Equation 5* inherits a number of properties from $C'_{ij}(0)$ in *Equation 4*: the second addend still remains the dominant one for $T_{ij} \lesssim \sigma$; inherited are also the absence of a weight change for fully overlapping fields ($\Delta w_{ij} = 0$ for $T_{ij} = 0$), the maximum weight change for $T_{ij} \approx \sqrt{2}\sigma$, and $\Delta w_{ij} \to 0$ for $T_{ij} \to \infty$ (*Figure 3A*). Furthermore, the prefactor $A^2\mu\tau^2$ in *Equation 5* suggests that the average weight change increases with increasing width $\tau$ of the learning window, but we emphasize that this increase is restricted to $\tau \ll 1/\omega$ (as we assumed for the derivation), which prohibits a generalization of the quadratic scaling to large $\tau$; the exact dependence on $\tau$ will be explained later.

To quantify how much better a sequence can be learned with phase precession as compared to phase locking, we use the ratio of the weight change $\Delta w_{ij}$ with phase precession ($c > 0$) and the weight change $\Delta w_{ij}(c = 0)$ without phase precession (*Figure 3A*), and define the benefit $B$ of phase precession as

$$B := \frac{\Delta w_{ij}}{\Delta w_{ij}(c = 0)} - 1. \tag{6}$$

By inserting *Equation 5* in *Equation 6*, we can explicitly calculate the benefit $B$ of phase precession (see *Equation 20* in Materials and methods and solid line in *Figure 3B*). For $T_{ij} \lesssim \sigma$ and $\omega^4\tau^4 \ll 1$

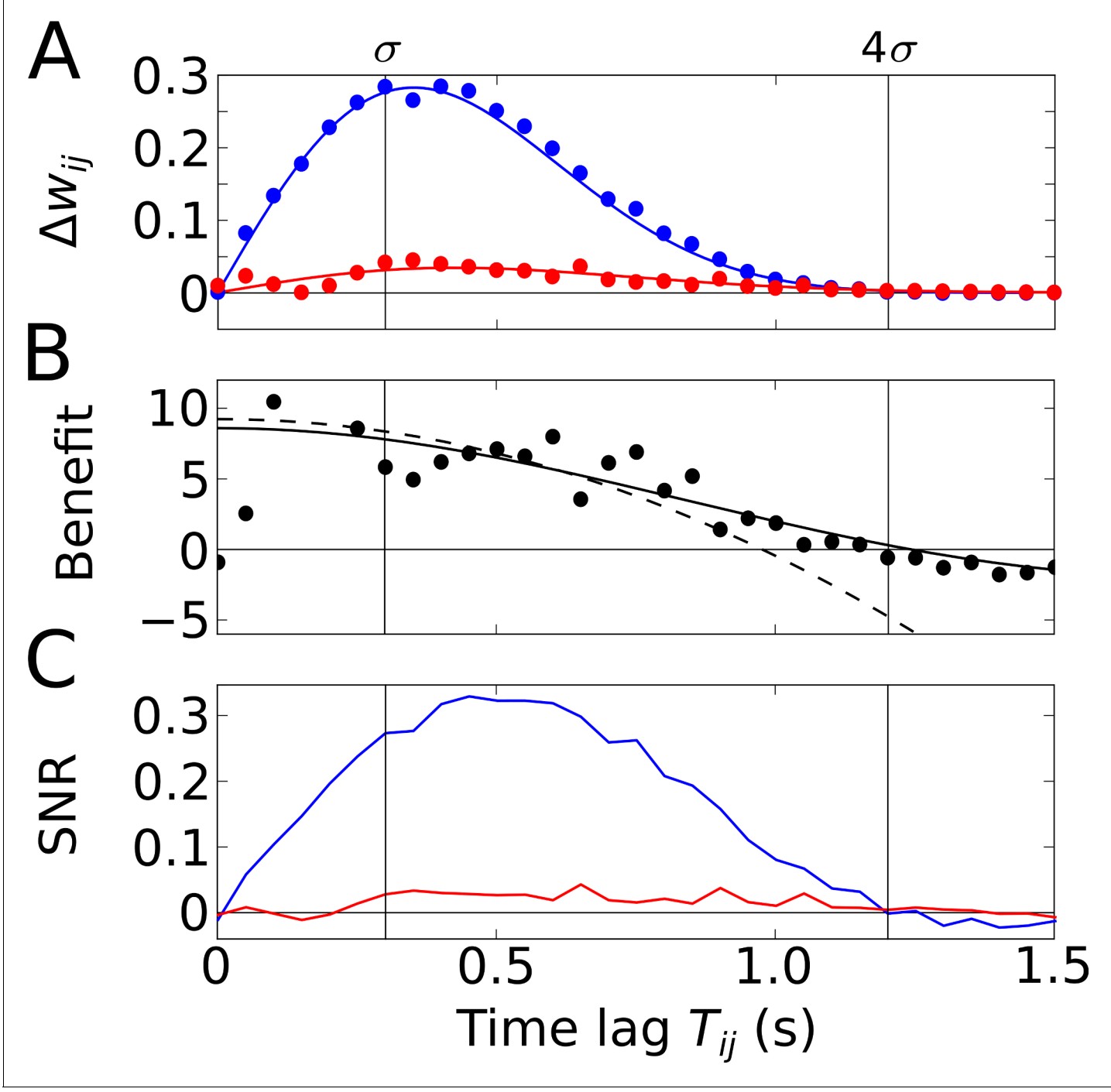

**Figure 3.** Temporal-order learning for narrow learning windows ($\tau \ll \frac{1}{\omega}$). (**A**) The average synaptic weight change $\Delta w_{ij}$ depends on the temporal separation $T_{ij}$ between the firing fields. Phase precession (blue) yields higher weight changes than phase locking (red). Simulation results (circles, averaged across $10^4$ repetitions) and analytical results (lines, *Equation 5*) match well. The vertical lines mark time lags of $\sigma$ and $4\sigma$, respectively, where $4\sigma$ approximates the total field width. (**B**) The benefit $B$ of phase precession is determined by the ratio of the average weight changes of two scenarios from (**A**). The solid and dashed lines depict the analytical expression for the benefit (*Equation 20*) and its approximation for small $T_{ij}$ (*Equation 7*), respectively. (**C**) Signal-to-noise ratio (SNR) of the weight change as a function of the firing-field separation $T_{ij}$. The SNR is defined as the mean weight change divided by the standard deviation across trials in the simulation. Colors as in (**A**). Parameters for all plots: $\omega = 2\pi \cdot 10$ Hz, $\sigma = 0.3$ s, $\tau = 10$ ms, $\mu = 1$, $c = 0.042$, $A = 10$.

(see Materials and methods) the benefit $B$ is well approximated by a Taylor expansion up to third order in $T_{ij}$ (dashed line in *Figure 3B*),

$$B \approx \frac{2}{3}\omega^2\sigma^2 c\left[1 - \left(\frac{c}{4\sigma^2}\cdot\frac{1-\omega^2\tau^2}{1+\omega^2\tau^2}+\frac{1}{6}\omega^2 c^2\right)T_{ij}^2\right]. \tag{7}$$

The maximum of $B$ as a function of $T_{ij}$ is obtained for $T_{ij}=0$ (fully overlapping fields), but the average weight change $\Delta w_{ij}$ is zero at this point. We note, however, that $B$ decays slowly with increasing $T_{ij}$, so $B(T_{ij}=0)$ can be used to approximate the benefit for small field separations $T_{ij}$ (i.e. largely overlapping fields). For narrow ($\omega\tau \ll 1$) odd STDP windows and slope-size matching ($\omega\sigma c = \pi/4$), we find the maximum $B_{max} \approx \omega\sigma/2$, which has an interesting interpretation: If we relate $\sigma$ to the field size $L$ of a Gaussian firing field through $L = 4\sigma$ and if we relate the frequency $\omega$ to the period $T_\theta$ of a theta oscillation cycle through $T_\theta = 2\pi/\omega$, we obtain $B_{max} \approx 0.82\, L/T_\theta$, that is, the maximum benefit of phase precession is about the number of theta oscillation cycles in a firing field. The example in *Figure 3B* (with firing fields in *Figure 2A*) has the maximum benefit $B_{max} \approx 10$ and the benefit remains in this range for partly overlapping firing fields ($0 < T_{ij} \lesssim \sigma$). We thus conclude that phase precession can boost temporal-order learning by about an order of magnitude for typical cases in which learning windows are narrower than a theta oscillation cycle and overlapping firing fields are an order of magnitude wider than a theta oscillation cycle.

So far, we have considered 'average' weight changes that resulted from neural activity that was described by a deterministic firing rate. However, neural activity often shows large variability, that is, different traversals of the same firing field typically lead to very different spike trains. To account for such variability, we have simulated neural activity as inhomogeneous Poisson processes (see Materials and methods for details). As a result, the change of the weight of a synapse, which depends on the correlation between spikes of the presynaptic and the postsynaptic cells, is a stochastic variable. It is important to consider the variability of the weight change ('noise') in order to assess the significance of the average weight change. For this reason, we utilize the signal-to-noise ratio (SNR), that is, the mean weight change divided by its standard deviation (see Materials and methods for details). To do so, we perform stochastic simulations of spiking neurons and calculate the average weight change and its variability across trials. This is done for phase-precessing as well as phase-locked activity. To connect this approach to our previous results, we confirm that the average weight changes estimated from many fields traversals matches well the analytical predictions (*Figure 3A and B*, see Materials and methods for details).

The SNR shown in *Figure 3C* summarizes how reliable is the learning signal in a single traversal of the two firing fields — for the assumed odd learning window. The SNR further depends on $T_{ij}$ and follows a similar shape as the weight changes in *Figure 3A*. For phase precession, there is a maximum SNR that is slightly shifted to larger $T_{ij}$; for phase locking, SNR is always much lower. For the synapse connecting two cells with firing fields as in *Figure 2A* where $T_{ij} = \sigma$, we find an SNR of 0.27, which is insufficient for a reliable representation of a sequence.

To allow reliable temporal-order learning, one possible solution is to increase the number of spikes per field traversal $A$ ($\text{SNR} \propto \sqrt{A}$, as shown in Appendix 1). Another possibility is to increase the number of synapses. In Materials and methods we show that $\text{SNR} \propto \sqrt{M}$ where $M$ is the number of identical and uncorrelated synapses. Therefore, to achieve $\text{SNR} \geq 1$ for $A = 10$, one needs $M \geq 14$ synapses.

In summary, for narrow, odd learning windows ($\tau \ll 1/\omega \ll \sigma$), temporal-order learning could benefit tremendously from phase precession as long as firing fields have some overlap. Average weight changes and the SNR are highest, however, for clearly distinct but still overlapping firing fields. It should be noted that any even component of the learning window would increase the noise and thus further decrease the SNR.

## Dependence of temporal-order learning on the width of the learning window for overlapping firing fields

To investigate how temporal-order learning for an odd learning window depends on its width, we vary the parameter $\tau$ and quantify the average synaptic weight change $\Delta w_{ij}$ and the SNR both analytically and numerically. We first study overlapping firing fields (*Figure 4*) and later consider non-overlapping firing fields (*Figure 5*).

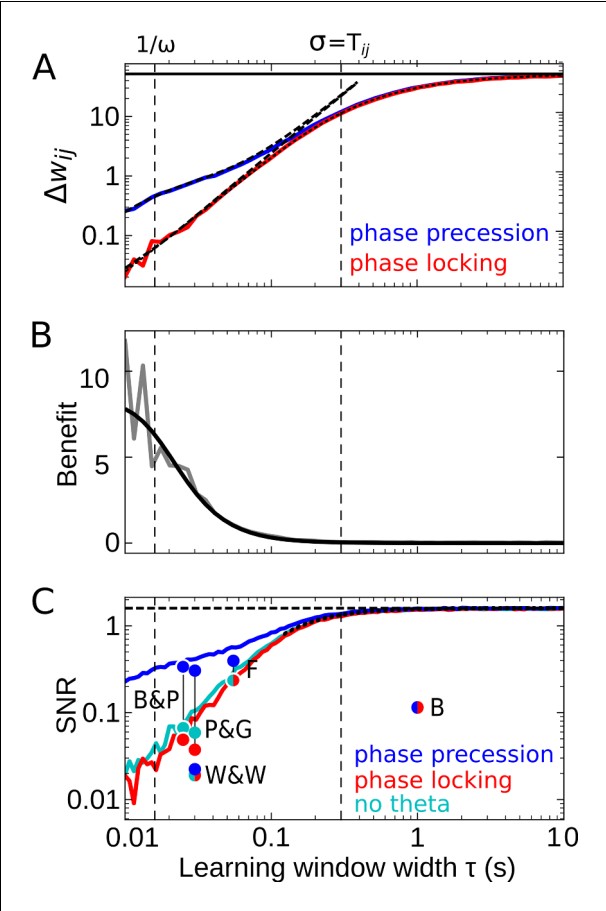

**Figure 4.** Effect of the learning-window width on temporal-order learning for overlapping fields (here: $T_{ij} = \sigma$). (**A**) Average weight change $\Delta w_{ij}$ as a function of width $\tau$ (for the asymmetric window $W$ in *Equation 14*) for phase precession and phase locking (colored curves). The solid black line depicts the theoretical maximum for large $\tau$ ($\Delta w_{ij} \approx 52$, *Equation 8*). The dashed curves show the analytical small-tau approximations (*Equation 5*). The dotted curve depicts the analytical approximation for the 'no theta' case (*Equation A2-46* in Appendix 2). The vertical dashed lines mark $1/\omega \approx 0.016$ s and the value of $\sigma = T_{ij} = 0.3$s, respectively. (**B**) The benefit $B$ of phase precession is largest for narrow learning windows, and it approaches 0 for wide windows. Simulations (gray line) and analytical result (black line, small-tau approximation from *Equation 20*) match well. (**C**) The signal-to-noise ratio (SNR; phase precession: blue, phase locking: red, no theta: cyan) takes into account that only the asymmetric part of the learning window is helpful for temporal-order learning. For large $\tau$, all three coding scenarios induce the same SNR. The horizontal dashed black line depicts the analytical limit of the SNR for large $\tau$ and overlapping firing fields (SNR $\approx 1.6$, *Equation A1-17* of Appendix 1). The dotted black line depicts the analytical expression for the 'no theta' case (*Equation A2-48* in Appendix 2, the curve could not be plotted for $\tau \lesssim 0.1$ s due to numerical instabilities). Dots represent the SNR for experimentally observed learning windows. The learning windows were taken from 'B&P', *Bi and Poo, 2001*: their Figure 1, 'F', *Froemke et al., 2005*: their Figure 1D bottom, 'W&W', *Wittenberg and Wang, 2006*: their Figure 3, 'P&G', *Pfister and Gerstner, 2006*: their Table 4, 'All to All', 'minimal model', and 'B', *Bittner et al., 2017*: their Figure 3D. For 'B&P', 'F', and 'B', the position of the dots on the horizontal axis was estimated as the average time constants for positive and negative lobes of the learning windows. Wittenberg and Wang modeled their learning rule by a difference of Gaussians — we approximated the corresponding time constant as 30 ms. For the triplet rule by Pfister and Gerstner, we used the average of three time constants: the two pairwise-interaction time constants (as in Bi and Poo) and the triplet-potentiation time constant. Parameters for all plots: $T_{ij} = 0.3$ s, $\omega = 2\pi \cdot 10$ Hz, $\sigma = 0.3$ s, $c = 0.042$, $A = 10$, $\mu = 1$. Colored/gray curves and dots are obtained from stochastic simulations; see Materials and methods for details.

For partly overlapping firing fields (e.g. $T_{ij} = \sigma$), we find numerically that the average synaptic weight change $\Delta w_{ij}$ (the 'learning signal') increases monotonically for increasing $\tau$ and saturates (colored curves in *Figure 4A*). This is because for increasing $\tau$ the overlap between the learning window

and the cross-correlation function grows, and this overlap begins to saturate as soon as the learning window is wider than $T_{ij}$, that is, the value at which the cross-correlation assumes its maximum (cmp. dashed curve in *Figure 2C*). To analytically calculate the saturation value of $\Delta w_{ij}$ for large learning-window widths ($\tau \gg \sigma$), we can approximate the learning window as a step function (see Materials and methods for details) and find the maximum

$$\Delta w_{ij}^{\max} \approx A^2 \mu \operatorname{erf}\left(\frac{T_{ij}}{2\sigma}\right) \tag{8}$$

that provides an upper bound to the weight change for overlapping firing fields (solid line in *Figure 4A*). For $\tau \lesssim 1/\omega$ (and actually well beyond this region), the analytical small-tau approximation of $\Delta w_{ij}$ (*Equation 5*, dashed curves in *Figure 4A*) matches the numerical results well.

The results in *Figure 4A* confirm that $\Delta w_{ij}$ is increased by phase precession for narrow learning windows but is independent of phase precession for $\tau \gg 1/\omega$. Thus, the benefit $B$ becomes small for large $\tau$ (*Figure 4B*) because, for large enough $\tau$, the theta oscillation completes multiple cycles within the width of the learning window. To better understand this behavior, let us return to *Equation 1*: if the product of a *wide* learning window and the cross-correlation $C_{ij}$ is integrated to obtain the weight change, the oscillatory modulation of the cross-correlation (e.g. as in *Figure 2C*) becomes irrelevant; similarly, according to *Equation 3*, the particular value of the derivative $C'_{ij}(t)$ near $t = 0$ can be neglected. Consequently, for $\tau \gg 1/\omega$ phase precession and phase locking as well as the scenario of firing fields that are not theta modulated yield the same weight change (*Figure 4A*), and the benefit approaches 0 (*Figure 4B*). Wide learning windows thus ignore the temporal (theta) fine-structure of the cross-correlation.

How noisy is this learning signal $\Delta w_{ij}$ across trials? *Figure 4C* shows that for odd learning windows the SNR increases with increasing $\tau$ and, for $\tau \gg \frac{1}{\omega}$, approaches a constant value. This constant value is the same for phase precession, phase locking, or no theta oscillations at all. Taken together, for large enough $\tau$, the advantage of phase precession vanishes. For small enough $\tau$, phase precession increases the SNR, which confirms and generalizes the results in *Figure 3C*. Remarkably, the SNR for 'phase locking' is lower than the one for 'no theta', which means that theta oscillations without phase precession degrade temporal-order learning, even though theta oscillations as such were emphasized to improve the modification of synaptic strength in many other cases (e.g. *Buzsáki, 2002*; *D'Albis et al., 2015*).

*Figure 4C* predicts that a large $\tau$ yields the biggest SNR, and thus wide learning windows are the best choice for temporal-order learning; however, we note that this conclusion is restricted to odd (i.e. asymmetric) learning windows. An additional even (i.e. symmetric) component of a learning window would increase the noise without affecting the signal, and thus would decrease the SNR (dots in *Figure 4C*). It is remarkable that the only experimentally observed instance of a wide window (with $\tau \approx 1$ s in *Bittner et al., 2017*) has a strong symmetric component, which leads to a low SNR (dot marked 'B' in *Figure 4C*).

Taken together, we predict that temporal-order learning would strongly benefit from wide, asymmetric windows. However, to date, all experimentally observed (predominantly) asymmetric windows are narrow (e.g. *Bi and Poo, 2001*; *Froemke et al., 2005*; *Wittenberg and Wang, 2006*; see *Abbott and Nelson, 2000*; *Bi and Poo, 2001* for reviews).

## Temporal-order learning for wide learning windows ($K \gg \sigma$)

We finally restrict our analysis to wide learning windows, which allows us then to also consider non-overlapping firing fields (*Figure 5A*, we again use two Gaussians with widths $\sigma$ and separation $T_{ij}$). To allow for temporal-order learning in this case, the spikes of two non-overlapping fields can only be 'paired' by a wide enough learning window. As already indicated in *Figure 4*, phase precession does not affect the weight change for such wide learning windows where the width $\tau$ of the learning window obeys $\tau \gg 1/\omega$ (note that we always assumed many theta oscillation cycles within a firing field, that is, $1/\omega \gg \sigma$). Furthermore, *Figure 4* indicated that only the asymmetric part of the learning window contributes to temporal-order learning. For the analysis of temporal-order learning with non-overlapping firing fields and wide learning windows, we thus ignore any theta modulation and phase precession and evaluate, again, only the odd STDP window $W(t) = \mu \operatorname{sign}(t) \exp(-|t|/\tau)$. In this case, the weight change (*Equation 1*) is still determined by the cross-correlation function and the

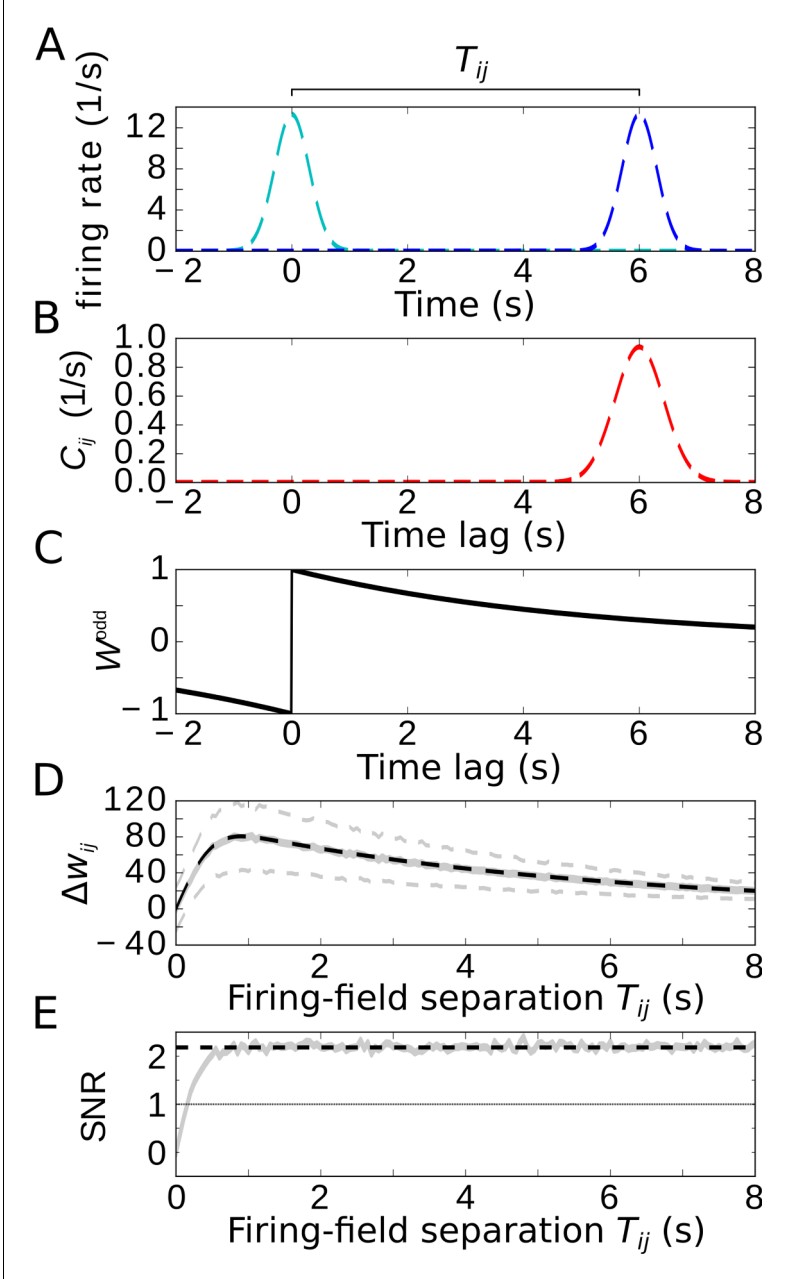

**Figure 5.** Temporal-order learning for non-overlapping firing fields using wide, asymmetric learning windows. (A) Firing rates of two example cells with non-overlapping firing fields. (B) Cross-correlation $C_{ij}$ of the two cells from (A). (C) Asymmetric learning window with large width ($\tau = 5$ s). (D) Resulting weight change $\Delta w_{ij}$ for wide learning window and non-overlapping firing fields. The solid gray line depicts the average weight change. The dashed gray lines represent ±1 standard deviation across 1000 repetitions of stochastic spiking simulations. The analytical curve (dashed black line, *Equation 9*) matches the simulation results. (E) SNR of the weight change. Results of the stochastic simulations are shown by the gray curve. The SNR saturates for larger $T_{ij}$, which fits the analytical expectation (dashed black line, *Equation 10*). Parameters, unless varied in a plot: $T_{ij} = 6$ s, $\sigma = 0.3$ s, $\tau = 5$ s, $\mu = 1$, $A = 10$.

learning window (examples in *Figure 5B,C*). The resulting weight change $\Delta w_{ij}$ as a function of the temporal separation $T_{ij}$ of firing fields is shown in *Figure 5D*: with increasing $T_{ij}$, the weight $\Delta w_{ij}$ quickly increases, reaches a maximum, and slowly decreases. The initial increase is due to the increasing overlap of the Gaussian bump in $C_{ij}$ with the positive lobe of the learning window. The

decrease, on the other hand, is dictated by the time course of the learning window. For $\tau \gg \sigma$, these two effects can be approximated by

$$\Delta w_{ij} \approx A^2 \mu \, \mathrm{erf}\left(\frac{T_{ij}}{2\sigma}\right) \exp\left(-\frac{T_{ij}}{\tau}\right) \qquad (9)$$

in which the error function describes the overlap of the cross-correlation with the learning window and the exponential term describes the decay of the learning window (dashed black curve in *Figure 5D*, see also *Equation 25* in Materials and methods for details).

How does the SNR of the weight change depend on the separation $T_{ij}$ of firing fields? For $T_{ij} = 0$, the signal is zero and thus also the SNR. As $T_{ij}$ increases, both signal and noise increase, but quickly settle on a constant ratio. The value of the SNR height of this plateau can be approximated by

$$\mathrm{SNR} \approx \frac{A}{\sqrt{2A+1}} \qquad (10)$$

(dashed line in *Figure 5E*), where $A$ is the number of spikes within a firing field (*Equation 11*). For $A = 10$, we find $\mathrm{SNR} \approx 2.2$, allowing for temporal-order learning with a single synapse. We note that this conclusion is limited to asymmetric STDP windows. A symmetric component (like in *Bittner et al., 2017*) decreases the SNR and makes temporal-order learning less efficient.

Taken together, temporal-order learning can be performed with wide STDP windows, and phase precession does not provide any benefit; but temporal-order learning requires a purely asymmetric plasticity window. For non-overlapping firing fields, wide learning windows are essential to bridge a temporal gap between the fields.

## Discussion

In this report, we show that phase precession facilitates the learning of the temporal order of behavioral sequences for asymmetric learning windows that are shorter than a theta cycle. To quantify this improvement, we use additive, pairwise STDP and calculate the expected weight change for synapses between two activated cells in a sequence. We confirm the long-held hypothesis (*Skaggs et al., 1996*) that phase precession bridges the vastly different time scales of the slow sequence of behavioral events and the fast STDP rule. Synaptic weight changes can be an order of magnitude higher when phase precession organizes the spiking of multiple cells at the theta time scale as compared to phase-locking cells.

### Other mechanisms and models for sequence learning

As an alternative mechanism to bridge the time scales of behavioral events and the induction of synaptic plasticity, *Drew and Abbott, 2006* suggested STDP and persistent activity of neurons that code for such events. The authors assume regularly firing neurons that slowly decrease their firing rate after the event and show that this leads to a temporal compression of the sequence of behavioral events. For stochastically firing neurons, this approach is similar to ours with two overlapping, unmodulated Gaussian firing fields. In this case, sequence learning is possible, but the efficiency can be improved considerably by phase precession.

*Sato and Yamaguchi, 2003* as well as *Shen et al., 2007* investigated the memory storage of behavioral sequences using phase precession and STDP in a network model. In computer simulations, they find that phase precession facilitates sequence learning, which is in line with our results. In contrast to these approaches, our study focuses on a minimal network (two cells), but this simplification allows us to (i) consider a biologically plausible implementation of STDP, firing fields, and phase precession and (ii) derive analytical results. These mathematical results predict parameter dependencies, which is difficult to achieve with only computer simulations.

Related to our work is also the approach by Masquelier and colleagues *Masquelier et al., 2009* who showed that pattern detection can be performed by single neurons using STDP and phase coding, yet they did not include phase precession. They consider patterns in the input whereas, in our framework, it might be argued that patterns between input and output are detected instead.

## Noisy activity of neurons and prediction of the minimum number of synapses for temporal-order learning

To account for stochastic spiking, we use Poisson neurons. We find that a single synapse is not sufficient to reliably encode a minimal two-neuron sequence in a single trial because the fluctuations of the weight change are too large. Fortunately, the SNR scales with $\sqrt{MA}$, that is, the square root of the number $M$ of identical, but independent synapses and the number $A$ of spikes per field traversal of the neurons. For generic hippocampal place fields and typical STDP, we predict that about 14 synapses are sufficient to reliably encode temporal order in a single traversal. Interestingly, peak firing rates of place fields are remarkably high (up to 50 spikes/s; e.g. *O'Keefe and Recce, 1993*, *Huxter et al., 2003*). Taken together, in hippocampal networks, reliable encoding of the temporal order of a sequence is possible with a low number of synapses, which matches simulation results on memory replay (*Chenkov et al., 2017*).

## Width, shape, and symmetry of the STDP window are critical for temporal-order learning

Various widths have been observed for STDP learning windows (*Abbott and Nelson, 2000*; *Bi and Poo, 2001*). We show that for all experimentally found STDP time constants phase precession can improve temporal-order learning. However, for learning windows much wider than a theta oscillation cycle, the benefit of phase precession for temporal-order learning is small. Wide learning windows, where the width can be even on a behavioral time scale of $\approx 1$ s (*Bittner et al., 2017*) or larger, could, on the other hand, enable the association of non-overlapping firing fields. Alternatively, non-overlapping firing fields might also be associated by narrow learning windows if additional cells (with firing fields that fill the temporal gap) help to bridge a large temporal difference, much like 'time cells' in the hippocampal formation (reviewed in *Eichenbaum, 2014*).

STDP windows typically have symmetric and asymmetric components (*Abbott and Nelson, 2000*; *Mishra et al., 2016*). We find that only the asymmetric component supports the learning of temporal order. In contrast, the symmetric component strengthens both forward and backward synapses by the same amount and thus contributes to the association of behavioral events independent of their temporal order. For example, the learning window reported by *Bittner et al., 2017* shows only a mild asymmetry and is thus unfavorable to store the temporal order of behavioral events. Only long, predominantly asymmetric STDP windows would allow for effective temporal-order learning (*Figure 4*).

Generally, the shape of STDP windows is subject to neuromodulation; for example, cholinergic and adrenergic modulation can alter its polarity and symmetry (*Hasselmo, 1999*). Also dopamine can change the symmetry of the learning window (*Zhang et al., 2009*). Therefore, sequence learning could be modulated by the behavioral state (attention, reward, etc.) of the animal.

## Key features of phase precession for temporal order-learning: generalization to non-periodic modulation of activity

For STDP windows narrower ($\leq 10$ ms) than a theta cycle ($\geq 100$ ms), we argue that the slope of the cross-correlation function at zero offset controls the change of the weight of the synapse connecting two neurons; and we show that phase precession can substantially increase this slope. This result predicts that features of the cross-correlation at temporal offsets that are larger than the width of the learning window are irrelevant for temporal-order learning. It is thus conceivable to boost temporal-order learning even without phase precession, which is weak if theta oscillations are weak, as for example in bats (*Ulanovsky and Moss, 2007*) and humans (*Herweg and Kahana, 2018*; *Qasim et al., 2021*). In this case, temporal-order learning may instead benefit from two other phenomena that could create an appropriate shape of the cross-correlation: (i) Spiking of cells is locked to common (aperiodic) fluctuations of excitability. (ii) Each cell responds the faster to an increase in its excitability the longer ago its firing field has been entered, which may be mediated by a progressive facilitation mechanism. Together, these phenomena can make the cross-correlation exhibit a steeper slope around zero and could even give rise to a local maximum at a positive offset. This temporal fine structure is superimposed on a slower modulation, which is related to the widths of the firing fields. In summary, a progressively decreasing delay of spiking with respect to non-rhythmic fluctuations in excitation generalizes the notion of phase precession. Interestingly, synaptic

short-term facilitation, which could generate the described fine structure of the cross-correlation, has also been proposed as mechanism underlying phase precession (*Leibold et al., 2008*).

## Model assumptions

In our model, we assumed that recurrent synapses (e.g. between neurons representing a sequence) are plastic but weak during encoding, such that they have a negligible influence on the postsynaptic firing rate; and that the feedforward input dominates neuronal activity. These assumptions seem justified as *Hasselmo, 1999* indicated that excitatory feedback connections may be suppressed during encoding to avoid interference from previously stored information (see also *Haam et al., 2018*). Furthermore, neuromodulators facilitate long-term plasticity (reviewed, e.g. by *Rebola et al., 2017*), which also supports our assumptions.

The assumption of weak recurrent connections implies that these connections do not affect the dynamics. Consequently (and in contrast to *Tsodyks et al., 1996*), we thus hypothesize that phase precession is not generated by the local, recurrent network (see also, e.g. *Chadwick et al., 2016*); instead, we assume that phase precession is inherited from upstream feedforward inputs (*Chance, 2012*; *Jaramillo et al., 2014*) or generated locally by a cellular/synaptic mechanism (*Magee, 2001*; *Harris et al., 2002*; *Mehta et al., 2002*; *Thurley et al., 2008*). After temporal-order learning was successful, the resulting asymmetric connections could indeed also generate phase precession (as demonstrated by the simulations in *Tsodyks et al., 1996*), and this phase precession could then even be similar to the one that has initially helped to shape synaptic connections. Finally, inherited or local cellularly/synaptically generated phase precession and locally network-generated phase precession could interact (as reviewed, for example in *Jaramillo and Kempter, 2017*).

We assumed in our model that the widths of the two firing fields that represent two events in a sequence are identical (see, e.g. *Figure 2A*). But firing fields may have different widths, and in this case a slope-size matched phase precession would fail to reproduce the timing of spikes required for the learning of the correct temporal order of the two events. For example, the learned temporal order of events (timed according to field entry) would even be reversed if two fields with different sizes are aligned at their ends. How could the correct temporal order nevertheless be learned in our framework? In the hippocampus, theta oscillations are a traveling wave (*Lubenov and Siapas, 2009*; *Patel et al., 2012*) such that there is a positive phase offset of theta oscillations for the wider firing fields in the more ventral parts of the hippocampus. This traveling-wave phenomenon could preserve the temporal order in the phase-precession-induced compressed spike timing, as also pointed out earlier (*Leibold and Monsalve-Mercado, 2017*; *Muller et al., 2018*).

Our results on learning rules for sequence learning rely on pairwise STDP in which pairs of presynaptic and postsynaptic spikes are considered. Conversely, triplet STDP considers also motifs of three spikes (either 2 presynaptic - 1 postsynaptic or 2 postsynaptic - 1 presynaptic) (*Pfister and Gerstner, 2006*). Triplets STDP models can reproduce a number of experimental findings that pairwise STDP could not, for example the dependence on the repetition frequency of spike pairs (*Sjöström et al., 2001*). To investigate the influence of triplet interactions on sequence learning, we implemented the generic triplet rule by *Pfister and Gerstner, 2006*. We used their 'minimal' model, which was regarded as the best model in terms of number of free parameters and fitting error; for the parameters they obtained from fitting the triplet STDP model to hippocampal data, we found only mild differences to our results (see, e.g. *Figure 4C*). Differences are small because the fitted time constant of the triplet term (40 ms) is smaller than typical inter-spike intervals ($\gtrsim 50$ ms, minimum in field centers) in our simulations.

## Replay of sequences and storage of multiple and overlapping sequences

A sequence imprinted in recurrent synaptic weights can be replayed during rest or sleep (*Wilson and McNaughton, 1994*; *Nádasdy et al., 1999*; *Diba and Buzsáki, 2007*; *Peyrache et al., 2009*; *Davidson et al., 2009*), which was also observed in network-simulation studies (*Matheus Gauy et al., 2020*; *Malerba and Bazhenov, 2019*; *Gillett et al., 2020*). Replay could thus be a possible readout of the temporal-order learning mechanism. However, replay depends on the many parameters of the network, and a thorough investigation of is beyond the scope of this

manuscript. Therefore, we focus on synaptic weight changes that represent the formation of sequences in the network, which underlies replay, and we do not simulate replay.

We have considered the minimal example of a sequence of two neurons. Sequences can contain many more neurons, and the question arises how two different sequences can be told apart if they both contain a certain neuron, but proceed in different directions — as they might do for sequences of spatial or non-spatial events (*Wood et al., 2000*). In this case, it may be beneficial to not only strengthen synapses that connect direct successors in the sequence but also synapses that connect the second-to-next neuron. In this way, the two crossing sequences could be disambiguated, and the wider context in which an event is embedded becomes associated, which is in line with retrieved-context theories of serial-order memory (*Long and Kahana, 2019*). More generally, it is an interesting question of how many sequences can be stored in a network of a given size. *Gillett et al., 2020* were able to analytically calculate the storage capacity for the storage of sequences in a Hebbian network.

In conclusion, our model predicts that phase precession enables efficient and robust temporal-order learning. To test this hypothesis, we suggest experiments that modulate the shape of the STDP window or selectively manipulate phase precession and evaluate memory of temporal order.

# Materials and methods

## Experimental design: model description

We model the time-dependent firing rate of a phase precessing cell $i$ (two examples in *Figure 2A*) as

$$f_i(t) = A \cdot G_{\mu_i,\sigma}(t) \cdot \{1 + \cos[\omega(t - c\mu_i)]\}, \tag{11}$$

where the scaling factor $A$ determines the number of spikes per field traversal and $G_{\mu_i,\sigma}(t) = 1/(\sqrt{2\pi}\sigma) \cdot \exp[-(t - \mu_i)^2/(2\sigma^2)]$ is a Gaussian function that describes a firing field with center at $\mu_i$ and width $\sigma$. The firing field is sinusoidally modulated with theta frequency $\omega$ (but the sinusoidal modulation is not a critical assumption, see *Discussion*), with typically many oscillation cycles in a firing field ($\omega\sigma \gg 1$). The compression factor $c$ can be used to vary between phase precession ($c > 0$), phase locking ($c = 0$), and phase recession ($c < 0$) because the average population activity of many such cells oscillates at frequency of $(1 - c)\omega$ (*Geisler et al., 2010*; *D'Albis et al., 2015*), which provides a reference frame to assign theta phases (*Figure 2A*). Usually, $|c| \ll 1$ with typical values $c \lesssim 1/(\sigma\omega)$ (*Geisler et al., 2010*); for a pair of cells with overlapping firing fields (centers separated by $T_{ij} := \mu_j - \mu_i$) the phase delay is $\omega c T_{ij}$ (*Figure 2B*).

To quantify temporal-order learning, we consider the average weight change $\Delta w_{ij}$ of the synapse from cell $i$ to cell $j$, which is (*Kempter et al., 1999*)

$$\Delta w_{ij} = \int_{-\infty}^{\infty} dt \, W(t) \, C_{ij}(t) \tag{12}$$

where $C_{ij}(t)$ is the cross-correlation between the firing rates $f_i$ and $f_j$ of cells $i$ and $j$, respectively (*Figure 2C,D*):

$$C_{ij}(t) = \int_{-\infty}^{\infty} dt' f_i(t') f_j(t + t'). \tag{13}$$

$W(t)$ denotes the synaptic learning window, for example the asymmetric window

$$W(t) = \mu \begin{cases} +\exp(-t/\tau), & t \geq 0 \\ -\exp(+t/\tau), & t < 0, \end{cases} \tag{14}$$

where $\tau$ is the time constant and $\mu > 0$ is the learning rate (*Figure 2E*).

For the following calculations, we make two assumptions that are reasonable in the hippocampal formation (*O'Keefe and Recce, 1993*; *Bi and Poo, 2001*; *Geisler et al., 2010*) :

1. The theta oscillation has multiple cycles within the Gaussian envelope of the firing field in *Equation 11* ($1/\omega \ll \sigma$).

2. The window $W$ is short compared to the theta period ($\tau \ll 1/\omega$).

## Analytical approximation of the cross-correlation function

To explicitly calculate the cross-correlation $C_{ij}(t)$ as defined in *Equation 13*, we plug in the firing-rate functions (*Equation 11*) for the two neurons:

$$
\begin{aligned}
C_{ij}(t) &= \int_{-\infty}^{\infty} \mathrm{d}t'\, A \cdot G_{0,\sigma}(t') \cdot [1 + \cos(\omega t')] \cdot A \cdot G_{T_{ij},\sigma}(t+t') \cdot \left\{ 1 + \cos\left[\omega(t+t'-cT_{ij})\right] \right\} \\
&= A^2 \int_{-\infty}^{\infty} \mathrm{d}t' \Big\{ G_{0,\sigma}(t') G_{T_{ij},\sigma}(t+t') \\
&\qquad + G_{0,\sigma}(t') G_{T_{ij},\sigma}(t+t') \cos(\omega t') \\
&\qquad + G_{0,\sigma}(t') G_{T_{ij},\sigma}(t+t') \cos\left[\omega(t+t'-cT_{ij})\right] \\
&\qquad + G_{0,\sigma}(t') G_{T_{ij},\sigma}(t+t') \cos(\omega t') \cos\left[\omega(t+t'-cT_{ij})\right] \Big\}.
\end{aligned}
$$

The first term (out of four) describes the cross-correlation of two Gaussians, which results in a Gaussian function centered at $T_{ij}$ and with width $\sigma\sqrt{2}$. For the second term, we note that the product of two Gaussians yields a function proportional to a Gaussian with width $\sigma/\sqrt{2}$, and then use assumption (i). When integrated, the second term's contribution to $C_{ij}(t)$ is negligible because the cosine function oscillates multiple times within the Gaussian bump, that is, positive and negative contributions to the integral approximately cancel. The same argument applies to the third term. For the fourth term, we use the trigonometric property $\cos(\alpha) \cdot \cos(\beta) = \frac{1}{2}(\cos(\alpha+\beta) + \cos(\alpha-\beta))$. We set $\alpha = \omega t'$, $\beta = \omega(t+t'-cT_{ij})$ and find

$$
\begin{aligned}
& G_{0,\sigma}(t') G_{T_{ij},\sigma}(t+t') \cos(\omega t') \cos\left[\omega(t+t'-cT_{ij})\right] \\
=\ & \frac{1}{2} G_{0,\sigma}(t') G_{T_{ij},\sigma}(t+t') \cos\left[\omega(t+2t'-cT_{ij})\right] + \frac{1}{2} G_{0,\sigma}(t') G_{T_{ij},\sigma}(t+t') \cos\left[\omega(t-cT_{ij})\right].
\end{aligned}
$$

Again, we use assumption (i) and neglect the first addend on the right-hand side. Notably, the cosine function in the second addend is independent of the integration variable $t'$. Taken together, we find

$$
C_{ij}(t) \approx A^2\, G_{T_{ij},\sigma\sqrt{2}}(t) \left\{ 1 + \frac{1}{2}\cos\left[\omega(t-cT_{ij})\right] \right\}. \tag{15}
$$

Thus, the cross-correlation can be approximated by a Gaussian function (center at $T_{ij}$, width $\sigma\sqrt{2}$) that is theta modulated with an amplitude scaled by the factor $\frac{1}{2}$.

To further simplify *Equation 15*, we note that the time constant $\tau$ of the STDP window is usually small compared to the theta period (assumption (ii), *Figure 2C,D,E*). Structures in $C_{ij}(t)$ for $|t| \gg \tau$ thus have a negligible effect on the synaptic weight change. Therefore, we can focus on the cross-correlation for small temporal lags. In this range, we approximate the (slow) Gaussian modulation of $C_{ij}(t)$ (*Figure 2C,D*, dashed red line) by a linear function, that is,

$$
\begin{aligned}
G_{T_{ij},\sqrt{2}\sigma}(t) &\approx \left.\frac{\mathrm{d}}{\mathrm{d}t} G_{T_{ij},\sqrt{2}\sigma}(t)\right|_{t=0} \cdot t + G_{T_{ij},\sqrt{2}\sigma}(0) \\
&= G_{T_{ij},\sqrt{2}\sigma}(0) \left( \frac{T_{ij}}{2\sigma^2} \cdot t + 1 \right).
\end{aligned} \tag{16}
$$

Inserting this result in *Equation 15*, we approximate the cross-correlation function $C_{ij}(t)$ for $|t| \lesssim \tau$ as (*Figure 2D*, dashed black line)

$$
C_{ij}(t) \approx A^2\, G_{T_{ij},\sqrt{2}\sigma}(0) \left( \frac{T_{ij}}{2\sigma^2} \cdot t + 1 \right) \left[ 1 + \frac{1}{2}\cos\left(\omega(t-cT_{ij})\right) \right]. \tag{17}
$$

In the *Results*, we show that the slope of the cross-correlation function at $t = 0$ is important for temporal-order learning. From *Equation 17* we find

$$C'_{ij}(0) \approx \frac{A^2 T_{ij}}{2\sigma^2} G_{T_{ij},\sqrt{2}\sigma}(0) \left[ 1 + \omega\sigma\,\omega c\sigma \frac{\sin(\omega c T_{ij})}{\omega c T_{ij}} + \frac{\cos(\omega c T_{ij})}{2} \right], \tag{18}$$

which has three addends within the square brackets. Let us estimate the relative size of the second and third terms with respect to the first one. The third term is at most of the order of 0.5 because $|\cos(\omega c T_{ij})| \leq 1$. For the second addend, we note that $\sin(\omega c T_{ij})/(\omega c T_{ij})$ approaches 1 for $T_{ij} \to 0$ and remains in this range for $|\omega c T_{ij}| \lesssim \pi/4$. This condition is fulfilled for $|T_{ij}| \lesssim \sigma$ if we assume slope-size matching of phase precession (*Geisler et al., 2010*), that is, $\omega c \sigma \approx \frac{\pi}{4} \approx 0.79$. Then, the size of the second addend is dictated by the factor $\omega\sigma$, which is large according to assumption (i). In other words, for typical phase precession and $|T_{ij}| \lesssim \sigma$, the second addend is much larger than the other two.

To further understand the structure of $C'_{ij}(0)$, which is also shaped by the prefactors in front of the square brackets, we first note that $C'_{ij}(0)$ is zero for fully overlapping firing fields ($T_{ij} \to 0$). On the other hand, for very large field separations ($T_{ij} \gg \sigma$), the Gaussian term $G$ causes $C'_{ij}(0)$ to become zero. The prefactors have a maximum at $|T_{ij}| = \sqrt{2}\sigma$. The maximum's exact location is slightly shifted by the second addend but remains near $\sqrt{2}\sigma$. This peak will be important because it is inherited by the average weight change (*Equation 3*).

## Average weight change

Having approximated the cross-correlation function and its slope at zero (*Equations 17,18*), we are now ready to calculate the average synaptic weight change (*Equation 3*) for the assumed STDP window (*Equation 14*). Standard integration methods yield

$$\Delta w_{ij} = \frac{A^2 \mu \tau^2 T_{ij}}{\sigma^2} G_{T_{ij},\sqrt{2}\sigma}(0) \left[ 1 + \frac{\omega^2\sigma^2 c}{\omega^2\tau^2 + 1} \frac{\sin(\omega c T_{ij})}{\omega c T_{ij}} + \frac{(1 - \omega^2\tau^2)\cos(\omega c T_{ij})}{2(1 + \omega^2\tau^2)^2} \right]. \tag{19}$$

Because $\Delta w_{ij}$ is a temporal average of $C'_{ij}(t)$ for small $t$ (see interpretation of *Equation 3*), the weight change's structure resembles the previously discussed structure of $C'_{ij}(0)$. The averaging introduces additional factors proportional to $1 \pm \omega^2\tau^2$, but for $\omega\tau \ll 1$ [assumption (ii)] those have only minor effects on the relative size of the three addends. The second term still dominates. Importantly, $\Delta w_{ij} = 0$ for $T_{ij} = 0$ and the position of the peak at $T_{ij} \lesssim \sqrt{2}\sigma$ is inherited from $C'_{ij}(0)$ (*Figure 3A*).

## The benefit of phase precession

To quantify the benefit $B$ of phase precession, we consider the expression $\Delta w_{ij}/\Delta w_{ij}(c = 0) - 1$, because $\Delta w_{ij}$ describes the overall weight change (including phase precession), and $\Delta w_{ij}(c = 0)$ serves as the baseline weight change due to the temporal separation of the firing fields (without phase precession). We subtract 1 to obtain $B = 0$ when the weight changes are the same with and without phase precession. From *Equation 19* we find

$$B = \frac{2}{3}\omega^2\sigma^2 c \cdot \frac{\sin(\omega c T_{ij})}{\omega c T_{ij}} \cdot \frac{1 + \omega^2\tau^2}{1 + \omega^2\tau^2 + \frac{2}{3}\omega^4\tau^4} + \frac{\cos(\omega c T_{ij}) - 1}{3} \cdot \frac{1 - \omega^2\tau^2}{1 + \omega^2\tau^2 + \frac{2}{3}\omega^4\tau^4}. \tag{20}$$

To better understand the structure of $B$, we Taylor-expand it in $T_{ij}$ up to the third order and assume $\omega^4\tau^4 \ll 1$ [assumption (ii)]. The result is

$$B \approx \frac{2}{3}\omega^2\sigma^2 c \left[ 1 - \left( \frac{1}{6}\omega^2 c^2 + \frac{c}{4\sigma^2} \cdot \frac{1 - \omega^2\tau^2}{1 + \omega^2\tau^2} \right) T_{ij}^2 \right]. \tag{21}$$

Thus, $B$ assumes a maximum for $T_{ij} = 0$ and slowly decays for small $T_{ij}$ (*Figure 3B*). Using slope-size matching ($\omega\sigma c = \pi/4$), the maximal benefit is

$$B_{\max} \approx \frac{\pi}{6}\omega\sigma = \frac{\pi^2}{12}\frac{L}{T_\theta} \approx 0.82\frac{L}{T_\theta}, \tag{22}$$

where $L = 4\sigma$ depicts the total field size and $T_\theta = \frac{2\pi}{\omega}$ is the period of the theta oscillation. Thus, the

number of theta cycles per firing field determines the benefit for small separations of the firing fields.

## Average weight change for wide learning windows

In this paragraph we relax assumption (ii), that is, we consider wide asymmetric learning windows $W$ (*Equation 14* with $\tau \gg \sigma$). Furthermore, we neglect any theta-oscillatory modulation of the firing fields in *Equation 11* and, thus, $C_{ij}$ in *Equation 15*.

First, for non-overlapping fields ($T_{ij} \gg \sigma$), the learning window can be approximated to be constant near the peak of the Gaussian bump of $C_{ij}$. We can thus rewrite *Equation 1* as

$$\Delta w_{ij} \approx W(T_{ij}) \int_{-\infty}^{\infty} C_{ij}(t)\, \mathrm{d}t = A^2 \mu \exp\left(-\frac{T_{ij}}{\tau}\right). \tag{23}$$

Second, for overlapping fields ($0 < T_{ij} \lesssim \sigma$), the Gaussian bump of $C_{ij}$ partly lies on the negative lobe of $W$. We can approximate $W(t) = \mathrm{sign}(t)$, and the average weight change in *Equation 1* then reads

$$\Delta w_{ij} \approx A^2 \mu \, \mathrm{erf}\left(\frac{T_{ij}}{2\sigma}\right). \tag{24}$$

Combining the two limiting cases in *Equations 23 and 24* yields

$$\Delta w_{ij} \approx A^2 \mu \, \mathrm{erf}\left(\frac{T_{ij}}{2\sigma}\right) \exp\left(-\frac{T_{ij}}{\tau}\right). \tag{25}$$

## Signal-to-noise ratio

To correctly encode the temporal order of behavioral events, the average weight change $\Delta w_{ij}$ of a forward synapse needs to be larger than the average weight change $\Delta w_{ji}$ of the corresponding backward synapse. We thus define the signal-to-noise ratio as

$$\mathrm{SNR} = \frac{\Delta w_{ij} - \Delta w_{ji}}{\mathrm{std}\left(\Delta w_{ij}^k\right) + \mathrm{std}\left(\Delta w_{ji}^k\right)},$$

where std() denotes the standard deviation and $\Delta w_{ij}^k$, $\Delta w_{ji}^k$ are the weight changes for trial $k \in [1,N]$, the averages across trials being $\Delta w_{ij} = \langle \Delta w_{ij}^k \rangle_k$ and $\Delta w_{ji} = \langle \Delta w_{ji}^k \rangle_k$. This expression for the SNR 'punishes' the non-sequence-specific strengthening of backward synapses. Specifically, $\mathrm{SNR} = 0$ for a symmetric (even) learning window, because the numerator (which represents the 'signal') is zero. On the other hand, a perfectly asymmetric learning window, like the one used throughout this study (*Equation 14*), yields $\mathrm{SNR} = \frac{\Delta w_{ij}}{\mathrm{std}(\Delta w_{ij}^k)}$, because $\Delta w_{ij}^k = -\Delta w_{ji}^k$. Asymmetric learning windows thus recover the classical definition of the SNR as the ratio between the average weight change and the standard deviation of the weight change.

We note that the generalized definition above can be used to calculate the SNR for arbitrary windows, such as the learning window from *Bittner et al., 2017*, *Figure 4C*.

Assuming an asymmetric window and $M$ uncorrelated synapses with the same mean and variance of the weight change, we can write the signal-to-noise ratio as

$$\mathrm{SNR} = \frac{M \cdot \Delta w_{ij}}{\sqrt{\mathrm{var}\left(\sum_{k=1}^{M} \Delta w_{ij}^k\right)}} = \sqrt{M} \frac{\Delta w_{ij}}{\mathrm{std}(\Delta w_{ij}^k)},$$

because the variance of the sum can be decomposed into the sum of variances and covariances. All covariances are zero because synapses are uncorrelated. This leaves a sum of $M$ variances, which are identical. Therefore, the standard deviation, and consequently also the SNR, scale with $\sqrt{M}$.

## Numerical simulations

To numerically simulate the synaptic weight change, spikes were generated by inhomogeneous Poisson processes with rate functions according to *Equation 11*. For every spike pair, the contribution to

the weight change was calculated according to *Equation 14*. We repeated the simulations for $N = 10^4$ trials, and the mean weight change as well as the standard deviation across trials and the SNR were estimated. All simulations were implemented in Python 3.8 using the packages NumPy (RRID:SCR_008633) and SciPy (RRID:SCR_008058). Matplotlib (RRID:SCR_008624) was used for plotting; Inkscape (RRID:SCR_014479) was used for final adjustments to the Figures. The Python code is available at https://gitlab.com/e.reifenstein/synaptic-learning-rules-for-sequence-learning (*Reifenstein and Kempter, 2021*; copy archived at swh:1:rev: 157c347a735a090f591a2b77a71b90d7de65bca5).

## Acknowledgements

This work was funded by the Deutsche Forschungsgemeinschaft (DFG, German Research Foundation; Grants GRK 1589/2, SPP 1665, SFB 1315 - project-ID 327654276) and the German Federal Ministry for Education and Research (BMBF; Grant 01GQ1705). We thank Lukas Kunz, Natalie Schieferstein, Tiziano D'Albis, Paul Pfeiffer, and Adam Wilkins for helpful discussions and feedback on the manuscript. ETR and RK designed the research. ETR, IBK, and RK performed the research, wrote and discussed the manuscript. ETR, IBK, and RK declare no conflict of interest.

## Additional information

### Funding

| Funder | Grant reference number | Author |
| --- | --- | --- |
| Deutsche Forschungsgemeinschaft | 01GQ1705 | Richard Kempter |
| Deutsche Forschungsgemeinschaft | GRK 1589/2 | Richard Kempter |
| Deutsche Forschungsgemeinschaft | SPP 1665 | Richard Kempter |
| Deutsche Forschungsgemeinschaft | SFB 1315 | Richard Kempter |

The funders had no role in study design, data collection and interpretation, or the decision to submit the work for publication.

### Author contributions

Eric Torsten Reifenstein, Conceptualization, Software, Formal analysis, Funding acquisition, Validation, Investigation, Visualization, Methodology, Writing - original draft, Project administration, Writing - review and editing; Ikhwan Bin Khalid, Validation, Investigation, Methodology, Writing - review and editing; Richard Kempter, Conceptualization, Resources, Formal analysis, Supervision, Funding acquisition, Validation, Investigation, Visualization, Methodology, Writing - original draft, Project administration, Writing - review and editing

### Author ORCIDs

Eric Torsten Reifenstein (iD) https://orcid.org/0000-0002-6898-0178
Ikhwan Bin Khalid (iD) https://orcid.org/0000-0003-1783-2834
Richard Kempter (iD) https://orcid.org/0000-0002-5344-2983

### Decision letter and Author response

Decision letter https://doi.org/10.7554/eLife.67171.sa1
Author response https://doi.org/10.7554/eLife.67171.sa2

## Additional files

### Supplementary files

• Transparent reporting form

### Data availability

Code and data are available at https://gitlab.com/e.reifenstein/synaptic-learning-rules-for-sequence-learning (copy archived at https://archive.softwareheritage.org/swh:1:rev:157c347a735a090f591a2b77a71b90d7de65bca5).

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

# Appendix 1

## The signal-to-noise ratio of $\Delta w_{ij}$

A synapse with weight $w_{ij}$ is assumed to connect neuron $i$ to neuron $j$. Here, we aim to derive the signal-to-noise ratio (SNR) of the weight changes $\Delta w_{ij}$, which is defined as (Materials and methods)

$$\mathrm{SNR} = \frac{\langle \Delta w_{ij} \rangle - \langle \Delta w_{ji} \rangle}{\mathrm{std}\big(\Delta w_{ij}\big) + \mathrm{std}\big(\Delta w_{ji}\big)}. \tag{A1-1}$$

where $\langle \Delta w_{ij} \rangle$ is the average signal. The noise is described by the standard deviation of the weight change,

$$\mathrm{std}\big(\Delta w_{ij}\big) = \sqrt{\mathrm{var}(\Delta w_{ij})} = \sqrt{\langle \Delta w_{ij}^2 \rangle - \langle \Delta w_{ij} \rangle^2}.$$

Signal and noise are generated by additive STDP and spiking activity that is modeled by two inhomogeneous Poisson processes with rates $f_i(t)$ and $f_j(t)$ that have finite support. The average weight change is calculated by $\langle \Delta w_{ij} \rangle = \int \mathrm{d}t\, W(t) C_{ij}(t)$ where $W(t)$ is the synaptic learning window and $C_{ij}(t)$ depicts the cross-correlation function $C_{ij}(t) = \int \mathrm{d}t' f_i(t') f_j(t'+t)$. From ***Kempter et al., 1999***, we use

$$
\begin{aligned}
\langle \Delta w_{ij}^2 \rangle(t) = &-\Delta w_{ij}(t_0)^2 + 2\Delta w_{ij}(t_0)\langle \Delta w_{ij} \rangle(t) + \int_{t_0}^{t} \mathrm{d}t' \int_{t_0}^{t} \mathrm{d}u \\
&\left\{ \langle S_i(t')S_i(u) \rangle (w^{\mathrm{in}})^2 + \langle S_j(t')S_j(u) \rangle (w^{\mathrm{out}})^2 + \langle S_i(t')S_j(u) \rangle 2w^{\mathrm{in}}w^{\mathrm{out}} \right. \\
&+ 2\int \mathrm{d}s\, W(s)\big(\langle S_i(t')S_i(u+s)S_j(u) \rangle w^{\mathrm{in}} + \langle S_j(t')S_i(u+s)S_j(u) \rangle w^{\mathrm{out}}\big) \\
&\left. + \int \mathrm{d}s \int \mathrm{d}v\, W(s)W(v)\langle S_i(t'+s)S_i(u+v)S_j(t')S_j(u) \rangle \right\},
\end{aligned}
\tag{A1-2}
$$

where $S_i(t) = \sum_n \delta(t - t_i^{(n)})$ and $S_j(t) = \sum_n \delta(t - t_j^{(n)})$ are the presynaptic and postsynaptic spike trains, respectively. To simplify, we set $t_0 = -\infty$ and $\Delta w_{ij}(t_0) = 0$. Furthermore, we are interested in paired STDP and thus set $w^{\mathrm{in}} = w^{\mathrm{out}} = 0$. For $\langle \Delta w_{ij}^2 \rangle = \lim_{t \to \infty} \langle \Delta w_{ij}^2 \rangle(t)$, ***Equation A1-2*** reduces to

$$\langle \Delta w_{ij}^2 \rangle = \int_{-\infty}^{\infty} \mathrm{d}t' \int_{-\infty}^{\infty} \mathrm{d}u \int \mathrm{d}s \int \mathrm{d}v\, W(s)W(v)\langle S_i(t'+s)S_i(u+v)S_j(t')S_j(u) \rangle. \tag{A1-3}$$

Because both spike trains are drawn from different Poisson processes, $S_i$ and $S_j$ are statistically independent, and therefore we can simplify

$$\langle S_i(t'+s)S_i(u+v)S_j(t')S_j(u) \rangle = \langle S_i(t'+s)S_i(u+v) \rangle \langle S_j(t')S_j(u) \rangle.$$

Moreover, in a spike train the spikes at different times are uncorrelated,

$$\langle S_i(t'+s)S_i(u+v) \rangle = \langle S_i(t'+s) \rangle \langle S_i(u+v) \rangle + \langle S_i(t'+s) \rangle \delta(t'+s-u-v)$$

and

$$\langle S_j(t')S_j(u) \rangle = \langle S_j(t') \rangle \langle S_j(u) \rangle + \langle S_j(t') \rangle \delta(t'-u).$$

As $S_i$ and $S_j$ are realizations of inhomogeneous Poisson processes with rates $f_i(t)$ and $f_j(t)$, respectively, we find

$$\langle S_i(t'+s)S_i(u+v) \rangle = f_i(t'+s)f_i(u+v) + f_i(t'+s)\delta(t'+s-u-v)$$

and

$$\langle S_j(t')S_j(u) \rangle = f_j(t')f_j(u) + f_j(t')\delta(t'-u).$$

We insert these expressions into ***Equation A1-3***:

$$\langle \Delta w_{ij}^2 \rangle = \int_{-\infty}^{\infty} \mathrm{d}t' \int_{-\infty}^{\infty} \mathrm{d}u \int \mathrm{d}s \int \mathrm{d}v \, W(s)W(v)F(t',s,u,v) \tag{A1-4}$$

where

$$F(t',s,u,v) = \left[f_i(t'+s)f_i(u+v) + f_i(t'+s)\delta(t'+s-u-v)\right] \cdot \left[f_j(t')f_j(u) + f_j(t')\delta(t'-u)\right].$$

To explicitly calculate the SNR, we parameterize the firing rates as

$$f_i(t) = \frac{A}{\sqrt{2\pi}\sigma} \exp\left(-\frac{t^2}{2\sigma^2}\right) \cdot \left[1 + \cos(\omega t)\right] \tag{A1-5}$$

and

$$f_j(t) = \frac{A}{\sqrt{2\pi}\sigma} \exp\left(-\frac{(t - T_{ij})^2}{2\sigma^2}\right) \cdot \left[1 + \cos(\omega(t - cT_{ij}))\right]. \tag{A1-6}$$

See main text for definitions of symbols. Furthermore, we assume $W(s) = W^{\mathrm{odd}}(s) + W^{\mathrm{even}}(s)$ with

$$W^{\mathrm{odd}}(s) = \mu \begin{cases} +\exp(-s/\tau), & s \geq 0 \\ -\exp(+s/\tau), & s < 0, \end{cases}$$

(see *Equation 14* in Materials and methods) and

$$W^{\mathrm{even}} = \lambda \exp(-|s|/\kappa).$$

In what follows we consider a limiting case of wide learning windows, for which we can explicitly calculate the SNR. The results obtained in this case match well to the numerical simulations for wide learning windows (*Figures 4* and *5* in the main text).

## Wide learning windows

For wide windows (formally: $\tau \to \infty$, $\kappa \to \infty$), we can approximate $W^{\mathrm{even}} = \lambda$ and $W^{\mathrm{odd}}(t) = \mu \operatorname{sgn}(t)$ and neglect the sinusoidal modulations of $f_i$ and $f_j$ in *Equation A1-5 and A1-6*; phase precession does not affect the SNR in this case.

The following calculations are similar for odd and even windows. We elaborate the calculations in detail for odd windows and use '$\pm$' and '$\mp$' to include the similar calculations for even windows. The top symbol ('+' and '—', respectively) corresponds to odd windows; the bottom symbol corresponds to even windows.

To start, we split the third and fourth integral in *Equation A1-4* into positive and negative time lags $s$ and $v$, respectively:

$$\begin{aligned}
\frac{1}{\mu^2}\langle \Delta w_{ij}^2 \rangle(t) &= \int_{-\infty}^{\infty} \mathrm{d}t' \int_{-\infty}^{\infty} \mathrm{d}u \int \mathrm{d}s \int \mathrm{d}v \operatorname{sgn}(s)\operatorname{sgn}(v)F(t',s,u,v) \\
&= \int_{-\infty}^{\infty} \mathrm{d}t' \int_{-\infty}^{\infty} \mathrm{d}u \left\{ \int_0^{\infty} \mathrm{d}s \int_0^{\infty} \mathrm{d}v F(t',s,u,v) \right. \\
&\qquad \mp \int_0^{\infty} \mathrm{d}s \int_{-\infty}^{0} \mathrm{d}v F(t',s,u,v) \\
&\qquad \mp \int_{-\infty}^{0} \mathrm{d}s \int_0^{\infty} \mathrm{d}v F(t',s,u,v) \\
&\qquad \left. + \int_{-\infty}^{0} \mathrm{d}s \int_{-\infty}^{0} \mathrm{d}v F(t',s,u,v) \right\}
\end{aligned} \tag{A1-7}$$

We rewrite $F$ as

$$F(t',s,u,v) = \underbrace{f_i(t'+s)f_i(u+v)f_j(t')f_j(u)}_{(i)} + \underbrace{f_i(t'+s)f_i(u+v)f_j(t')\delta(t'-u)}_{(ii)} + \underbrace{f_i(t'+s)\delta(t'+s-u-v)f_j(t')f_j(u)}_{(iii)} + \underbrace{f_i(t'+s)\delta(t'+s-u-v)f_j(t')\delta(t'-u)}_{(iv)},$$

which has four addends and occurs in four integrals in *Equation A1-7*. Thus, there are 16 terms we need to evaluate. We label these terms (1.i) to (1.iv) for the first integral, (2.i) to (2.iv) for the second integral and so on until (4.iv).

For the term (1.i) we find

$$
\begin{aligned}
&\int_{-\infty}^{\infty} dt' \int_{-\infty}^{\infty} du \int_{0}^{\infty} ds \int_{0}^{\infty} dv f_i(t'+s) f_i(u+v) f_j(t') f_j(u) \\
&= \int_{-\infty}^{\infty} dt' f_j(t') \int_{-\infty}^{\infty} du f_j(u) \int_{0}^{\infty} ds f_i(t'+s) \int_{0}^{\infty} dv f_i(u+v) \\
&= \frac{A^2}{4} \int_{-\infty}^{\infty} dt' f_j(t') \left[1 - \mathrm{erf}\left(\frac{t'}{\sqrt{2}\sigma}\right)\right] \int_{-\infty}^{\infty} du f_j(u) \left[1 - \mathrm{erf}\left(\frac{u}{\sqrt{2}\sigma}\right)\right] \\
&= \frac{A^2}{4} \left\{ \int_{-\infty}^{\infty} dt' f_j(t') \left[1 - \mathrm{erf}\left(\frac{t'}{\sqrt{2}\sigma}\right)\right] \right\}^2 \\
&= \frac{A^2}{4} \left\{ \int_{-\infty}^{\infty} dt' f_j(t') - \int_{-\infty}^{\infty} dt' f_j(t') \mathrm{erf}\left(\frac{t'}{\sqrt{2}\sigma}\right) \right\}^2.
\end{aligned}
\tag{A1-8}
$$

The first integral is $\int_{-\infty}^{\infty} dt' f_j(t') = A$. The second integral can be solved by taking the derivative with respect to $T_{ij}$:

$$
\begin{aligned}
&\int_{-\infty}^{\infty} dt' f_j(t') \mathrm{erf}\left(\frac{t'}{\sqrt{2}\sigma}\right) \\
&= \int dT_{ij} \int_{-\infty}^{\infty} dt' \frac{d}{dT_{ij}} f_j(t') \mathrm{erf}\left(\frac{t'}{\sqrt{2}\sigma}\right) \\
&= \int dT_{ij} \int_{-\infty}^{\infty} dt' \left[-f_j'(t')\right] \mathrm{erf}\left(\frac{t'}{\sqrt{2}\sigma}\right) \\
&\quad \left(\text{because } \frac{d}{dT_{ij}} f_j(t') = -\frac{d}{dt'} f_j(t') \equiv -f_j'(t')\right) \\
&= \int dT_{ij} \left\{ \left[\mathrm{erf}\left(\frac{t'}{\sqrt{2}\sigma}\right)\left[-f_j(t')\right]\right]_{-\infty}^{\infty} - \int_{-\infty}^{\infty} dt' \frac{2}{A} f_i(t')\left[-f_j(t')\right] \right\} \\
&\quad (\text{integration by parts}) \\
&= \int dT_{ij} \left\{ \left[1\cdot 0 - (-1)\cdot 0\right] + \frac{2}{A}\cdot\frac{A^2}{2\sqrt{\pi}\sigma} \exp\left(-\frac{T_{ij}^2}{4\sigma^2}\right) \right\} \\
&= \frac{A}{\sqrt{\pi}\sigma} \int dT_{ij} \exp\left(-\frac{T_{ij}^2}{4\sigma^2}\right) \\
&= A\, \mathrm{erf}\left(\frac{T_{ij}}{2\sigma}\right)
\end{aligned}
\tag{A1-9}
$$

$$
\tag{A1-10}
$$

Term (1.i) (*Equation A1-8*) thus reads:

$$
\begin{aligned}
&\int_{-\infty}^{\infty} dt' \int_{-\infty}^{\infty} du \int_{0}^{\infty} ds \int_{0}^{\infty} dv f_i(t'+s) f_i(u+v) f_j(t') f_j(u) \\
&= \frac{A^2}{4}\left[A - A\,\mathrm{erf}\left(\frac{T_{ij}}{2\sigma}\right)\right]^2 = \frac{A^4}{4}\left[1 - \mathrm{erf}\left(\frac{T_{ij}}{2\sigma}\right)\right]^2
\end{aligned}
$$

For (2.i) we find

$$
\begin{aligned}
&\mp \int_{-\infty}^{\infty} dt' \int_{-\infty}^{\infty} du \int_{0}^{\infty} ds \int_{-\infty}^{0} dv f_i(t'+s) f_i(u+v) f_j(t') f_j(u) \\
&= \mp \int_{-\infty}^{\infty} dt' f_j(t') \int_{-\infty}^{\infty} du f_j(u) \int_{0}^{\infty} ds f_i(t'+s) \int_{-\infty}^{0} dv f_i(u+v) \\
&= \mp \frac{A^2}{4} \int_{-\infty}^{\infty} dt' f_j(t') \left[1 - \mathrm{erf}\left(\frac{t'}{\sqrt{2}\sigma}\right)\right] \int_{-\infty}^{\infty} du f_j(u) \left[1 + \mathrm{erf}\left(\frac{u}{\sqrt{2}\sigma}\right)\right] \\
&= \mp \frac{A^2}{4} \left[A - A\,\mathrm{erf}\left(\frac{T_{ij}}{2\sigma}\right)\right]\left[A + A\,\mathrm{erf}\left(\frac{T_{ij}}{2\sigma}\right)\right] \quad (\text{using Equation A1-10}) \\
&= \mp \frac{A^4}{4}\left[1 - \mathrm{erf}^2\left(\frac{T_{ij}}{2\sigma}\right)\right]
\end{aligned}
$$

Term (3.i) is symmetric to (2.i) and thus yields the same result. For (4.i) we find (in analogy to the term (1.i)):

$$
\int_{-\infty}^{\infty} dt' \int_{-\infty}^{\infty} du \int_{-\infty}^{0} ds \int_{-\infty}^{0} dv f_i(t'+s) f_i(u+v) f_j(t') f_j(u)
$$
$$
= \frac{A^2}{4} \left( \int_{-\infty}^{\infty} dt' f_j(t') \left[ 1 + \mathrm{erf}\left( \frac{t'}{\sqrt{2}\sigma} \right) \right] \right)^2
$$
$$
= \frac{A^4}{4} \left[ 1 + \mathrm{erf}\left( \frac{T_{ij}}{2\sigma} \right) \right]^2
$$

We sum the contributions (1.i) to (4.i) for the odd learning window:

$$
\frac{A^4}{4} \left[ 1 - \mathrm{erf}\left( \frac{T_{ij}}{2\sigma} \right) \right]^2 - 2 \cdot \frac{A^4}{4} \left[ 1 - \mathrm{erf}^2\left( \frac{T_{ij}}{2\sigma} \right) \right] + \frac{A^4}{4} \left[ 1 + \mathrm{erf}\left( \frac{T_{ij}}{2\sigma} \right) \right]^2
$$
$$
= \frac{A^4}{4} \left[ 1 - 2\,\mathrm{erf}\left( \frac{T_{ij}}{2\sigma} \right) + \mathrm{erf}^2\left( \frac{T_{ij}}{2\sigma} \right) - 2 + 2\,\mathrm{erf}^2\left( \frac{T_{ij}}{2\sigma} \right) + 1 + 2\,\mathrm{erf}\left( \frac{T_{ij}}{2\sigma} \right) + \mathrm{erf}^2\left( \frac{T_{ij}}{2\sigma} \right) \right] \qquad \text{(A1-11)}
$$
$$
= A^4 \,\mathrm{erf}^2\left( \frac{T_{ij}}{2\sigma} \right)
$$

Let us continue with the second term of $F$, which is labeled by '(ii)', and consider the first (of four) integrals in *Equation A1-7*, that is, we continue with contribution (1.ii):

$$
\int_{-\infty}^{\infty} dt' \int_{-\infty}^{\infty} du \int_{0}^{\infty} ds \int_{0}^{\infty} dv f_i(t'+s) f_i(u+v) f_j(t') \delta(t'-u)
$$
$$
= \int_{-\infty}^{\infty} dt' f_j(t') \int_{-\infty}^{\infty} du\, \delta(t'-u) \int_{0}^{\infty} ds f_i(t'+s) \int_{0}^{\infty} dv f_i(u+v)
$$
$$
= \frac{A^2}{4} \int_{-\infty}^{\infty} dt' f_j(t') \left[ 1 - \mathrm{erf}\left( \frac{t'}{\sqrt{2}\sigma} \right) \right] \int_{-\infty}^{\infty} du\, \delta(t'-u) \left[ 1 - \mathrm{erf}\left( \frac{u}{\sqrt{2}\sigma} \right) \right]
$$
$$
= \frac{A^2}{4} \int_{-\infty}^{\infty} dt' f_j(t') \left[ 1 - 2\,\mathrm{erf}\left( \frac{t'}{\sqrt{2}\sigma} \right) + \mathrm{erf}^2\left( \frac{t'}{\sqrt{2}\sigma} \right) \right]
$$
$$
= \frac{A^3}{4} \left[ 1 - 2\,\mathrm{erf}\left( \frac{T_{ij}}{2\sigma} \right) + \frac{C}{A} \right] \quad \text{(using Equation A1-10),}
$$

with

$$
C = \int_{-\infty}^{\infty} dt' f_j(t') \,\mathrm{erf}^2\left( \frac{t'}{\sqrt{2}\sigma} \right),
$$

which will be solved later for special cases. Note that $C$ depends on $T_{ij}$ because $f_j(t')$ depends on $T_{ij}$. For (2.ii) we find:

$$
\mp \int_{-\infty}^{\infty} dt' \int_{-\infty}^{\infty} du \int_{0}^{\infty} ds \int_{-\infty}^{0} dv f_i(t'+s) f_i(u+v) f_j(t') \delta(t'-u)
$$
$$
= \mp \frac{A^2}{4} \int_{-\infty}^{\infty} dt' f_j(t') \left[ 1 - \mathrm{erf}\left( \frac{t'}{\sqrt{2}\sigma} \right) \right] \left[ 1 + \mathrm{erf}\left( \frac{t'}{\sqrt{2}\sigma} \right) \right]
$$
$$
= \mp \frac{A^2}{4} \int_{-\infty}^{\infty} dt' f_j(t') \left[ 1 - \mathrm{erf}^2\left( \frac{t'}{\sqrt{2}\sigma} \right) \right]
$$
$$
= \mp \frac{A^3}{4} \left( 1 - \frac{C}{A} \right).
$$

For (3.ii) we find the same:

$$
\mp \int_{-\infty}^{\infty} dt' \int_{-\infty}^{\infty} du \int_{-\infty}^{0} ds \int_{0}^{\infty} dv f_i(t'+s) f_i(u+v) f_j(t') \delta(t'-u)
$$
$$
= \mp \frac{A^2}{4} \int_{-\infty}^{\infty} dt' f_j(t') \left[ 1 + \mathrm{erf}\left( \frac{t'}{\sqrt{2}\sigma} \right) \right] \left[ 1 - \mathrm{erf}\left( \frac{t'}{\sqrt{2}\sigma} \right) \right]
$$
$$
= \mp \frac{A^3}{4} \left( 1 - \frac{C}{A} \right).
$$

For (4.ii) we find:

$$\int_{-\infty}^{\infty} dt' \int_{-\infty}^{\infty} du \int_{-\infty}^{0} ds \int_{-\infty}^{0} dv f_i(t'+s) f_i(u+v) f_j(t') \delta(t'-u)$$

$$= \frac{A^2}{4} \int_{-\infty}^{\infty} dt' f_j(t') \left[1 + \mathrm{erf}\left(\frac{t'}{\sqrt{2}\sigma}\right)\right]^2$$

$$= \frac{A^2}{4} \int_{-\infty}^{\infty} dt' f_j(t') \left[1 + 2\,\mathrm{erf}\left(\frac{t'}{\sqrt{2}\sigma}\right) + \mathrm{erf}^2\left(\frac{t'}{\sqrt{2}\sigma}\right)\right]$$

$$= \frac{A^3}{4} \left[1 + 2\,\mathrm{erf}\left(\frac{T_{ij}}{2\sigma}\right) + \frac{C}{A}\right].$$

Summing contributions (1.ii) to (4.ii) for the odd window yields:

$$\frac{A^3}{4}\left[1 - 2\,\mathrm{erf}\left(\frac{T_{ij}}{2\sigma}\right) + \frac{C}{A}\right] - 2\cdot\frac{A^3}{4}\left(1 - \frac{C}{A}\right) + \frac{A^3}{4}\left[1 + 2\,\mathrm{erf}\left(\frac{T_{ij}}{2\sigma}\right) + \frac{C}{A}\right]$$

$$= \frac{A^3}{4}\cdot\frac{4C}{A} = CA^2 \tag{A1-12}$$

We continue with contribution (1.iii):

$$\int_{-\infty}^{\infty} dt' \int_{-\infty}^{\infty} du \int_{0}^{\infty} ds \int_{0}^{\infty} dv f_j(t') f_j(u) f_i(t'+s) \delta(t'+s-u-v)$$

$$= \int_{-\infty}^{\infty} dt' f_j(t') \int_{-\infty}^{\infty} du f_j(u) \int_{0}^{\infty} ds f_i(t'+s) \int_{0}^{\infty} dv \delta(t'+s-u-v)$$

Contribution (1.iii) is non-zero if the argument $t'+s-u-v$ of the delta function in the last integral (across $v$) is zero for some $v$, which varies from 0 to $\infty$. The argument of the delta function is thus zero for some $v$ if $0 \leq t'+s-u < \infty$, which we can rewrite as $u \leq t'+s$ and then use it in the integral across $u$, which leads to

$$\int_{-\infty}^{\infty} dt' f_j(t') \int_{0}^{\infty} ds f_i(t'+s) \int_{-\infty}^{t'+s} du f_j(u)$$

$$= \frac{A}{2} \int_{-\infty}^{\infty} dt' f_j(t') \int_{0}^{\infty} ds f_i(t'+s) \left[1 + \mathrm{erf}\left(\frac{s+t'-T_{ij}}{\sqrt{2}\sigma}\right)\right]$$

$$= \frac{A}{2} \int_{-\infty}^{\infty} dt' f_j(t') \left[\int_{0}^{\infty} ds f_i(t'+s) + \int_{0}^{\infty} ds f_i(t'+s)\,\mathrm{erf}\left(\frac{s+t'-T_{ij}}{\sqrt{2}\sigma}\right)\right]$$

$$= \frac{A}{2} \int_{-\infty}^{\infty} dt' f_j(t') \left[\frac{A}{2}\left(1 - \mathrm{erf}\left(\frac{t'}{\sqrt{2}\sigma}\right)\right) + \int_{0}^{\infty} ds f_i(t'+s)\,\mathrm{erf}\left(\frac{s+t'-T_{ij}}{\sqrt{2}\sigma}\right)\right]$$

$$= \frac{A^2}{4} \int_{-\infty}^{\infty} dt' f_j(t') \left[1 - \mathrm{erf}\left(\frac{t'}{\sqrt{2}\sigma}\right)\right] + \frac{A}{2} \int_{-\infty}^{\infty} dt' f_j(t') \int_{0}^{\infty} ds f_i(t'+s)\,\mathrm{erf}\left(\frac{s+t'-T_{ij}}{\sqrt{2}\sigma}\right)$$

$$= \frac{A^3}{4}\left[1 - \mathrm{erf}\left(\frac{T_{ij}}{2\sigma}\right)\right] + \frac{D\cdot A}{2} \quad \text{(using Equation A1-10)}$$

with $D := \int_{-\infty}^{\infty} dt' f_j(t') \int_{0}^{\infty} ds f_i(t'+s)\,\mathrm{erf}\left(\frac{s+t'-T_{ij}}{\sqrt{2}\sigma}\right)$. $D$ will be evaluated later for special cases.

Similarly to (1.iii), we treat (2.iii):

$$\mp \int_{-\infty}^{\infty} dt' \int_{-\infty}^{\infty} du \int_{0}^{\infty} ds \int_{-\infty}^{0} dv f_j(t') f_j(u) f_i(t'+s) \delta(t'+s-u-v)$$

$$= \mp \int_{-\infty}^{\infty} dt' f_j(t') \int_{0}^{\infty} ds f_i(t'+s) \int_{t'+s}^{\infty} du f_j(u)$$

$$= \mp \frac{A}{2} \int_{-\infty}^{\infty} dt' f_j(t') \int_{0}^{\infty} ds f_i(t'+s) \left[1 - \mathrm{erf}\left(\frac{s+t'-T_{ij}}{\sqrt{2}\sigma}\right)\right]$$

$$= \mp \frac{A}{2} \int_{-\infty}^{\infty} dt' f_j(t') \int_{0}^{\infty} ds f_i(t'+s) \pm \frac{A}{2} \int_{-\infty}^{\infty} dt' f_j(t') \int_{0}^{\infty} ds f_i(t'+s)\,\mathrm{erf}\left(\frac{s+t'-T_{ij}}{\sqrt{2}\sigma}\right)$$

$$= \mp \frac{A^2}{4} \int_{-\infty}^{\infty} dt' f_j(t') \left[1 - \mathrm{erf}\left(\frac{t'}{\sqrt{2}\sigma}\right)\right] \pm \frac{A}{2} \int_{-\infty}^{\infty} dt' f_j(t') \int_{0}^{\infty} ds f_i(t'+s)\,\mathrm{erf}\left(\frac{s+t'-T_{ij}}{\sqrt{2}\sigma}\right)$$

$$= \mp \frac{A^3}{4}\left[1 - \mathrm{erf}\left(\frac{T_{ij}}{2\sigma}\right)\right] \pm \frac{D\cdot A}{2}$$

For (3.iii) we find:

$$\mp \int_{-\infty}^{\infty} dt' \int_{-\infty}^{\infty} du \int_{-\infty}^{0} ds \int_{0}^{\infty} dv f_j(t') f_j(u) f_i(t'+s) \delta(t'+s-u-v)$$

$$= \mp \int_{-\infty}^{\infty} dt' f_j(t') \int_{-\infty}^{0} ds f_i(t'+s) \int_{-\infty}^{t'+s} du f_j(u)$$

$$= \mp \frac{A}{2} \int_{-\infty}^{\infty} dt' f_j(t') \int_{-\infty}^{0} ds f_i(t'+s) \left[1 + \mathrm{erf}\left(\frac{s+t'-T_{ij}}{\sqrt{2}\sigma}\right)\right]$$

$$= \mp \frac{A^2}{4} \int_{-\infty}^{\infty} dt' f_j(t') \left[1 + \mathrm{erf}\left(\frac{t'}{\sqrt{2}\sigma}\right)\right] \mp \frac{A}{2} \int_{-\infty}^{\infty} dt' f_j(t') \int_{-\infty}^{0} ds f_i(t'+s) \mathrm{erf}\left(\frac{s+t'-T_{ij}}{\sqrt{2}\sigma}\right)$$

$$= \mp \frac{A^3}{4} \left[1 + \mathrm{erf}\left(\frac{T_{ij}}{2\sigma}\right)\right] \mp \frac{D' \cdot A}{2}.$$

with $D' := \int_{-\infty}^{\infty} dt' f_j(t') \int_{-\infty}^{0} ds f_i(t'+s) \mathrm{erf}\left(\frac{s+t'-T_{ij}}{\sqrt{2}\sigma}\right)$, which we will evaluate later for special cases.

Finally, for (4.iii) we find

$$\int_{-\infty}^{\infty} dt' \int_{-\infty}^{\infty} du \int_{-\infty}^{0} ds \int_{-\infty}^{0} dv f_j(t') f_j(u) f_i(t'+s) \delta(t'+s-u-v)$$

$$= \int_{-\infty}^{\infty} dt' f_j(t') \int_{-\infty}^{0} ds f_i(t'+s) \int_{t'+s}^{\infty} du f_j(u)$$

$$= \frac{A}{2} \int_{-\infty}^{\infty} dt' f_j(t') \int_{-\infty}^{0} ds f_i(t'+s) \left[1 - \mathrm{erf}\left(\frac{s+t'-T_{ij}}{\sqrt{2}\sigma}\right)\right]$$

$$= \frac{A^2}{4} \int_{-\infty}^{\infty} dt' f_j(t') \left[1 + \mathrm{erf}\left(\frac{t'}{\sqrt{2}\sigma}\right)\right] - \frac{A}{2} \int_{-\infty}^{\infty} dt' f_j(t') \int_{-\infty}^{0} ds f_i(t'+s) \mathrm{erf}\left(\frac{s+t'-T_{ij}}{\sqrt{2}\sigma}\right)$$

$$= \frac{A^3}{4} \left[1 + \mathrm{erf}\left(\frac{T_{ij}}{2\sigma}\right)\right] - \frac{D' \cdot A}{2}.$$

To sum the four contributions (1.iii) to (4.iii) for the odd window, we note that the first terms (square brackets) of (1.iii) and (2.iii) cancel, as well as the first terms of (3.iii) and (4.iii). We thus obtain:

$$\frac{D \cdot A}{2} + \frac{D \cdot A}{2} - \frac{D' \cdot A}{2} - \frac{D' \cdot A}{2} = A(D - D').$$

We continue with contribution (1.iv):

$$\int_{-\infty}^{\infty} dt' \int_{-\infty}^{\infty} du \int_{0}^{\infty} ds \int_{0}^{\infty} dv f_i(t'+s) \delta(t'+s-u-v) f_j(t') \delta(t'-u)$$

$$= \int_{-\infty}^{\infty} dt' f_j(t') \int_{-\infty}^{\infty} du \delta(t'-u) \int_{0}^{\infty} ds f_i(t'+s) \int_{0}^{\infty} dv \delta(t'+s-u-v)$$

$$= \int_{-\infty}^{\infty} dt' f_j(t') \int_{0}^{\infty} ds f_i(t'+s) \int_{-\infty}^{t'+s} du \delta(t'-u)$$

$$= \int_{-\infty}^{\infty} dt' f_j(t') \int_{0}^{\infty} ds f_i(t'+s) \int_{-\infty}^{s} du' \delta(u') \quad (\text{with } u' = u - t')$$

$$= \int_{-\infty}^{\infty} dt' f_j(t') \int_{0}^{\infty} ds f_i(t'+s) \cdot \begin{cases} 1, & \text{for } s > 0 \\ 0, & \text{else} \end{cases}$$

$$= \int_{-\infty}^{\infty} dt' f_j(t') \int_{0}^{\infty} ds f_i(t'+s)$$

$$= \frac{A}{2} \int_{-\infty}^{\infty} dt' f_j(t') \left[1 - \mathrm{erf}\left(\frac{t'}{\sqrt{2}\sigma}\right)\right]$$

$$= \frac{A^2}{2} \left[1 - \mathrm{erf}\left(\frac{T_{ij}}{2\sigma}\right)\right] \quad (\text{using Equation A1-10})$$

By similar arguments, (2.iv) yields:

$$\mp \int_{-\infty}^{\infty} \mathrm{d}t' \int_{-\infty}^{\infty} \mathrm{d}u \int_{0}^{\infty} \mathrm{d}s \int_{-\infty}^{0} \mathrm{d}v f_i(t'+s)\delta(t'+s-u-v)f_j(t')\delta(t'-u)$$

$$= \mp \int_{-\infty}^{\infty} \mathrm{d}t' f_j(t') \int_{0}^{\infty} \mathrm{d}s f_i(t'+s) \int_{t'+s}^{\infty} \mathrm{d}u \, \delta(t'-u)$$

$$= \mp \int_{-\infty}^{\infty} \mathrm{d}t' f_j(t') \int_{0}^{\infty} \mathrm{d}s f_i(t'+s) \int_{s}^{\infty} \mathrm{d}u' \, \delta(u')$$

$$= \mp \int_{-\infty}^{\infty} \mathrm{d}t' f_j(t') \int_{0}^{\infty} \mathrm{d}s f_i(t'+s) \cdot \begin{cases} 1, & \text{for } s>0 \\ 0, & \text{else} \end{cases}$$

$$= 0 \, .$$

(3.iv) yields

$$\mp \int_{-\infty}^{\infty} \mathrm{d}t' \int_{-\infty}^{\infty} \mathrm{d}u \int_{-\infty}^{0} \mathrm{d}s \int_{0}^{\infty} \mathrm{d}v f_i(t'+s)\delta(t'+s-u-v)f_j(t')\delta(t'-u)$$

$$= \mp \int_{-\infty}^{\infty} \mathrm{d}t' f_j(t') \int_{-\infty}^{0} \mathrm{d}s f_i(t'+s) \int_{-\infty}^{t'+s} \mathrm{d}u \, \delta(t'-u)$$

$$= \mp \int_{-\infty}^{\infty} \mathrm{d}t' f_j(t') \int_{-\infty}^{0} \mathrm{d}s f_i(t'+s) \cdot \begin{cases} 1, & \text{for } s>0 \\ 0, & \text{else} \end{cases}$$

$$= 0 \, .$$

(4.iv) yields

$$\int_{-\infty}^{\infty} \mathrm{d}t' \int_{-\infty}^{\infty} \mathrm{d}u \int_{-\infty}^{0} \mathrm{d}s \int_{-\infty}^{0} \mathrm{d}v f_i(t'+s)\delta(t'+s-u-v)f_j(t')\delta(t'-u)$$

$$= \int_{-\infty}^{\infty} \mathrm{d}t' f_j(t') \int_{-\infty}^{0} \mathrm{d}s f_i(t'+s) \int_{t'+s}^{\infty} \mathrm{d}u \, \delta(t'-u)$$

$$= \int_{-\infty}^{\infty} \mathrm{d}t' f_j(t') \int_{-\infty}^{0} \mathrm{d}s f_i(t'+s) \cdot \begin{cases} 1, & \text{for } s>0 \\ 0, & \text{else} \end{cases}$$

$$= \int_{-\infty}^{\infty} \mathrm{d}t' f_j(t') \int_{-\infty}^{0} \mathrm{d}s f_i(t'+s)$$

$$= \frac{A}{2} \int_{-\infty}^{\infty} \mathrm{d}t' f_j(t') \left[ 1 + \mathrm{erf}\left(\frac{t'}{\sqrt{2}\sigma}\right) \right]$$

$$= \frac{A^2}{2} \left[ 1 + \mathrm{erf}\left(\frac{T_{ij}}{2\sigma}\right) \right] \quad \text{(using Equation A1-10)} \, .$$

We sum the contributions (1.iv) and (4.iv) and obtain $A^2$. We now collect all terms for the odd window:

$$\frac{1}{\mu^2}\langle \Delta w_{ij}^2 \rangle = A^4 \mathrm{erf}^2\left(\frac{T_{ij}}{2\sigma}\right) + CA^2 + A(D-D') + A^2 \, .$$

So far, we have calculated the second moment of $\Delta w_{ij}$. In order to determine the variance, we need to calculate the average weight change for the odd window:

$$
\begin{aligned}
\langle \Delta w_{ij} \rangle &= \int_{-\infty}^{\infty} dt\, W(t) C_{ij}(t) \\
&= \mu \int_{-\infty}^{\infty} dt\, \mathrm{sgn}(t) C_{ij}(t) \\
&= \mu \int_{-\infty}^{\infty} dt\, \mathrm{sgn}(t) \int_{-\infty}^{\infty} dt' f_i(t') f_j(t'+t) \\
&= \frac{A^2 \mu}{2\sqrt{\pi}\sigma} \int_{-\infty}^{\infty} dt\, \mathrm{sgn}(t) \exp\left(-\frac{(t-T_{ij})^2}{4\sigma^2}\right) \quad \text{(using Equation 15 of the main text)} \\
&= \frac{A^2 \mu}{2\sqrt{\pi}\sigma} \left[ \int_{0}^{\infty} dt\, \exp\left(-\frac{(t-T_{ij})^2}{4\sigma^2}\right) - \int_{-\infty}^{0} dt\, \exp\left(-\frac{(t-T_{ij})^2}{4\sigma^2}\right) \right] \\
&= \frac{A^2 \mu}{2} \left[ 1 + \mathrm{erf}\left(\frac{T_{ij}}{2\sigma}\right) - \left(1 - \mathrm{erf}\left(\frac{T_{ij}}{2\sigma}\right)\right) \right] \\
&= A^2 \mu\, \mathrm{erf}\left(\frac{T_{ij}}{2\sigma}\right).
\end{aligned}
\tag{A1-13}
$$

The variance thus reads:

$$
\begin{aligned}
\frac{1}{\mu^2} \mathrm{var}(\Delta w_{ij}) &= \langle \Delta w_{ij}^2 \rangle - \langle \Delta w_{ij} \rangle^2 \\
&= A^4 \mathrm{erf}^2\left(\frac{T_{ij}}{2\sigma}\right) + CA^2 + A(D-D') + A^2 - A^4 \mathrm{erf}^2\left(\frac{T_{ij}}{2\sigma}\right) \\
&= (C+1)A^2 + A(D-D').
\end{aligned}
\tag{A1-14}
$$

For the signal-to-noise ratio, we note that the definition from *Equation A1-1*, for odd learning windows, simplifies to

$$
\mathrm{SNR} = \frac{\langle \Delta w_{ij} \rangle - \langle \Delta w_{ji} \rangle}{\mathrm{std}(\Delta w_{ij}) + \mathrm{std}(\Delta w_{ji})} = \frac{\langle \Delta w_{ij} \rangle}{\mathrm{std}(\Delta w_{ij})},
$$

because $\Delta w_{ij} = -\Delta w_{ji}$ for odd learning windows.
We insert *Equation A1-13 and A1-14* and find

$$
\mathrm{SNR} = \frac{A^2 \, \mathrm{erf}\left(\frac{T_{ij}}{2\sigma}\right)}{\sqrt{(C+1)A^2 + A(D-D')}}.
$$

To obtain the final result, we have to evaluate $C$, $D$, and $D'$. We distinguish the two cases $T_{ij} \gg \sigma$ and $T_{ij} = \sigma$ to approximate these three terms:
1. $T_{ij} \gg \sigma$:

$$
\mathrm{erf}\left(\frac{T_{ij}}{2\sigma}\right) \approx 1
\tag{A1-15}
$$

$$
C = \int_{-\infty}^{\infty} dt' f_j(t') \mathrm{erf}^2\left(\frac{t'}{\sqrt{2}\sigma}\right) \approx A,
$$

because the Gaussian function $f_j$ is shifted far into the positive lobe of the error function.

$$D = \int_{-\infty}^{\infty} dt' f_j(t') \int_0^{\infty} ds f_i(t' + s) \operatorname{erf}\left(\frac{t' + s - T_{ij}}{\sqrt{2}\sigma}\right)$$

$$\approx \int_{-\infty}^{\infty} dt' f_j(t') \int_0^{\infty} ds [-f_i(t' + s)]$$

$$= -\frac{A}{2} \int_{-\infty}^{\infty} dt' f_j(t') \left[1 - \operatorname{erf}\left(\frac{t'}{\sqrt{2}\sigma}\right)\right]$$

$$= -\frac{A}{2} \left[A - A \operatorname{erf}\left(\frac{T_{ij}}{2\sigma}\right)\right] \quad \text{(using Equation A1-10)}$$

$$\approx -\frac{A}{2}[A - A] \quad \text{(using Equation A1-15)}$$

$$= 0.$$

$$D' = \int_{-\infty}^{\infty} dt' f_j(t') \int_{-\infty}^0 ds f_i(t' + s) \operatorname{erf}\left(\frac{t' + s - T_{ij}}{\sqrt{2}\sigma}\right)$$

$$= -\frac{A}{2} \int_{-\infty}^{\infty} dt' f_j(t') \left[1 + \operatorname{erf}\left(\frac{t'}{\sqrt{2}\sigma}\right)\right]$$

$$\approx -\frac{A}{2}[A + A] \quad \text{(using Equations A1-10 and A1-15)}$$

$$= -A^2.$$

Thus,

$$\mathrm{SNR} \approx \frac{A^2 \cdot 1}{\sqrt{(A+1)A^2 + A(0 + A^2)}} = \frac{A^2}{\sqrt{2A^3 + A^2}} = \frac{A}{\sqrt{2A + 1}}. \tag{A1-16}$$

This number (for $A = 10$) is indicated as the analytical comparison in *Figure 5E*. For large A, the SNR (*Equation A1-16*) approaches $\sqrt{A/2}$.

2. $T_{ij} = \sigma$:

$$\operatorname{erf}\left(\frac{T_{ij}}{2\sigma}\right) = \operatorname{erf}\left(\frac{1}{2}\right) \approx 0.52$$

$$C \approx 0.494 A,$$

$$D \approx -0.013 A^2,$$

$$D' \approx -0.507 A^2,$$

all of which we calculated numerically.

It follows:

$$\mathrm{SNR} \approx \frac{A^2 \operatorname{erf}\left(\frac{1}{2}\right)}{\sqrt{(C+1)A^2 + A(D - D')}}$$

$$\approx \frac{0.52 A^2}{\sqrt{0.494 A^3 + A^2 + 0.494 A^3}} \tag{A1-17}$$

$$\approx \frac{0.52 A^2}{\sqrt{0.99 A^3 + A^2}}$$

$$\stackrel{A=10}{\approx} 1.58$$

This number is plotted as the large-tau approximation in *Figure 4C*. For large $A$, we find $\mathrm{SNR} \propto \sqrt{A}$.

## Even windows

As argued in the main text, for even windows, the weight change $\Delta w_{ij}$ contains no information about the order of events because $\Delta w_{ij} = \Delta w_{ji}$. This can be seen from *Equation A1-1* of Appendix 1. The SNR is zero for purely even windows because the signal is zero. Nonetheless, we can calculate the

variance of the weight change. To do so, we collect all terms of $\langle \Delta w_{ij}^2 \rangle$ for even windows (indicated by the bottom symbol of all occurences of '$\pm$' and '$\mp$' in the previous section). Again, we assume wide windows ($\kappa \to \infty$).

Collecting the terms (1.i) to (4.i) yields

$$\frac{A^4}{4}\left[1 - \mathrm{erf}\left(\frac{T_{ij}}{2\sigma}\right)\right]^2 + 2 \cdot \frac{A^4}{4}\left[1 - \mathrm{erf}^2\left(\frac{T_{ij}}{2\sigma}\right)\right] + \frac{A^4}{4}\left[1 + \mathrm{erf}\left(\frac{T_{ij}}{2\sigma}\right)\right]^2 = A^4.$$

Similarly, we sum the terms (1.ii) to (4.ii):

$$\frac{A^3}{4}\left[1 - 2\,\mathrm{erf}\left(\frac{T_{ij}}{2\sigma}\right) + \frac{C}{A}\right] + 2 \cdot \frac{A^3}{4}\left(1 - \frac{C}{A}\right) + \frac{A^3}{4}\left[1 + 2\,\mathrm{erf}\left(\frac{T_{ij}}{2\sigma}\right) + \frac{C}{A}\right] = A^3.$$

We continue to collect the contributions (1.iii) to (4.iii):

$$\begin{aligned}
& \frac{A^3}{4}\left[1 - \mathrm{erf}\left(\frac{T_{ij}}{2\sigma}\right)\right] + \frac{D \cdot A}{2} \\
+\ & \frac{A^3}{4}\left[1 - \mathrm{erf}\left(\frac{T_{ij}}{2\sigma}\right)\right] - \frac{D \cdot A}{2} \\
+\ & \frac{A^3}{4}\left[1 + \mathrm{erf}\left(\frac{T_{ij}}{2\sigma}\right)\right] + \frac{D' \cdot A}{2} \\
+\ & \frac{A^3}{4}\left[1 + \mathrm{erf}\left(\frac{T_{ij}}{2\sigma}\right)\right] - \frac{D' \cdot A}{2} \\
=\ & A^3
\end{aligned}$$

Finally, summing (1.iv) to (4.iv) yields the same result as for the odd window: $A^2$.

Overall,

$$\frac{1}{\lambda^2}\langle \Delta w_{ij}^2 \rangle = \frac{1}{\lambda^2}\langle \Delta w_{ji}^2 \rangle = A^4 + 2A^3 + A^2.$$

Together with

$$\frac{1}{\lambda}\langle \Delta w_{ij} \rangle = \frac{1}{\lambda}\langle \Delta w_{ji} \rangle = A^2,$$

the variance reads:

$$\frac{1}{\lambda^2}\mathrm{var}(\Delta w_{ij}) = \frac{1}{\lambda^2}\mathrm{var}(\Delta w_{ji}) = 2A^3 + A^2.$$

We now insert these variances in the denominator of *Equation A1-1*:

$$\frac{1}{\lambda}\left[\mathrm{std}(\Delta w_{ij}) + \mathrm{std}(\Delta w_{ji})\right] = 2\sqrt{2A^3 + A^2},$$

which, assuming $\mu = \lambda$, is twice the noise as for odd windows ($\sqrt{2A^3 + A^2}$, *Equation A1-16*).

In summary, for a complex learning window with even and odd contributions, the signal solely depends on the odd part, whereas both parts, even and odd, contribute to the noise. Any even contribution thus only decreases the SNR.

## Appendix 2

### Calculating SNR for learning windows of arbitrary width

We again consider odd learning windows of the shape

$$W^{\mathrm{odd}}(s) = \mu \begin{cases} +\exp(-s/\tau), & s \ge 0 \\ -\exp(+s/\tau), & s < 0. \end{cases} \tag{A2-1}$$

As in the case of wide learning windows, we again consider the second moment of the weight change (similar to *Equation A1-4* of Appendix 1):

$$
\begin{aligned}
\frac{\langle \Delta w_{ij}^2 \rangle(t)}{\mu^2} &= \frac{1}{\mu^2} \int_{-\infty}^{\infty} \mathrm{d}t' \int_{-\infty}^{\infty} \mathrm{d}u \int \mathrm{d}s \int \mathrm{d}v W^{\mathrm{odd}}(s) W^{\mathrm{odd}}(v) F(t',s,u,v) \\
&= \int_{-\infty}^{\infty} \mathrm{d}t' \int_{-\infty}^{\infty} \mathrm{d}u \Bigg\{ \int_0^{\infty} \mathrm{d}s \int_0^{\infty} \mathrm{d}v \exp(-s/\tau) \exp(-v/\tau) F(t',s,u,v) \\
&\quad - \int_0^{\infty} \mathrm{d}s \int_{-\infty}^{0} \mathrm{d}v \exp(-s/\tau) \exp(v/\tau) F(t',s,u,v) \\
&\quad - \int_{-\infty}^{0} \mathrm{d}s \int_0^{\infty} \mathrm{d}v \exp(s/\tau) \exp(-v/\tau) F(t',s,u,v) \\
&\quad + \int_{-\infty}^{0} \mathrm{d}s \int_{-\infty}^{0} \mathrm{d}v \exp(s/\tau) \exp(v/\tau) F(t',s,u,v) \Bigg\}
\end{aligned}
\tag{A2-2}
$$

We write F similarly as before, neglecting the theta modulation of the firing rate:

$$
\begin{aligned}
F(t',s,u,v) =\ & f_i(t'+s) f_i(u+v) f_j(t') f_j(u) + f_i(t'+s) f_i(u+v) f_j(t') \delta(t'-u) + \\
& f_i(t'+s) \delta(t'+s-u-v) f_j(t') f_j(u) + f_i(t'+s) \delta(t'+s-u-v) f_j(t') \delta(t'-u)
\end{aligned}
\tag{A2-3}
$$

with

$$f_i(t) = \frac{A}{\sqrt{2\pi}\sigma} \exp\left(-\frac{t^2}{2\sigma^2}\right) \tag{A2-4}$$

and

$$f_j(t) = \frac{A}{\sqrt{2\pi}\sigma} \exp\left(-\frac{(t-T_{ij})^2}{2\sigma^2}\right). \tag{A2-5}$$

We label the addends of *Equation A2-2* as $\{1,2,3,4\}$ and the addends of *Equation A2-3* as $\{i,ii,iii,iv\}$. In evaluating the second moment of the weight change, we realize that many integrands have similar forms, that is, products of exponentials, error functions, and delta functions. Consequently, we will first state the integral identities we use, and will then explicitly derive the term $(2.iii)$ as an example. The other terms can be evaluated in a similar manner.

### Integral identities

For the evaluation of the second moment of the weight change, many integrands consist of exponential functions containing linear and squared terms. To tackle these integrals, we use *Albano et al., 2011*

$$\int_a^{\infty} \exp\left[-q^2 x^2 - px\right] \mathrm{d}x = \frac{\sqrt{\pi}}{2q} \exp\left[\frac{p^2}{4q^2}\right] \left[1 - \mathrm{erf}\left(\frac{p+2aq^2}{2q}\right)\right] \quad \text{with } q>0,\ a>0,\ p>0. \tag{A2-6}$$

The second recurring form of integrals is

$$\int_{-\infty}^{\infty} \mathrm{d}t\ \exp\left[-a^2(t^2+bt)\right] \mathrm{erf}(-a\{t+c\}) \quad \text{with } a>0 \text{ and } b,c \in \mathbb{R}. \tag{A2-7}$$

Substituting

$$x := -a(t+c) \qquad\qquad \mathrm{d}x = -a\,\mathrm{d}t$$
$$t := -c - \frac{x}{a} \qquad\qquad t^2 = c^2 + 2\frac{c}{a}x + \frac{x^2}{a^2},$$

we can rewrite

$$\int_{-\infty}^{\infty} \mathrm{d}t \, \exp\left[-a^2(t^2+bt)\right]\mathrm{erf}(-a\{t+c\}) = \frac{1}{a}\exp\left[\frac{a^2b^2}{4}\right]\int_{-\infty}^{\infty}\mathrm{d}x\,\exp\left[-\left(x+a\left\{c-\frac{b}{2}\right\}\right)^2\right]\mathrm{erf}[x]. \qquad \text{(A2-8)}$$

We now use an integral identity by *Ng and Geller, 1969* (their section 4.3, eq. 13):

$$\int_{-\infty}^{\infty}\mathrm{d}x\,\mathrm{erf}(x)\exp\left[-(px+q)^2\right]\mathrm{d}x = -\frac{\sqrt{\pi}}{p}\mathrm{erf}\left[\frac{q}{\sqrt{p^2+1}}\right], \qquad \text{(A2-9)}$$

which yields the desired solution ($p = 1, q = a\{c - b/2\}$):

$$\int_{-\infty}^{\infty}\mathrm{d}t\,\exp\left[-a^2(t^2+bt)\right]\mathrm{erf}(-a\{t+c\}) = -\frac{\sqrt{\pi}}{a}\exp\left[\frac{a^2b^2}{4}\right]\mathrm{erf}\left[\frac{a}{\sqrt{2}}\left(c-\frac{b}{2}\right)\right]. \qquad \text{(A2-10)}$$

## Example: deriving the term (2.iii)

For the term (2.*iii*), we have

$$
\begin{aligned}
(2.iii) &= -\int_{-\infty}^{\infty}\mathrm{d}t'\int_{-\infty}^{\infty}\mathrm{d}u\int_{0}^{\infty}\mathrm{d}s\int_{-\infty}^{0}\mathrm{d}v\,\exp\left[-\frac{s}{\tau}\right]\exp\left[\frac{v}{\tau}\right]f_i(t'+s)\delta(t'+s-u-v)f_j(t')f_j(u) \\
&= -\left(\frac{A}{\sqrt{2\pi}\sigma}\right)^3\int_{-\infty}^{\infty}\mathrm{d}t'\,\exp\left[-\frac{(t'-T_{ij})^2}{2\sigma^2}\right]\exp\left[-\frac{t'^2}{2\sigma^2}\right]\int_{-\infty}^{\infty}\mathrm{d}u\,\exp\left[-\frac{(u-T_{ij})^2}{2\sigma^2}\right] \\
&\quad \int_{0}^{\infty}\mathrm{d}s\,\exp\left[-\frac{1}{2\sigma^2}\left(s^2+2s\left\{t'+\frac{\sigma^2}{\tau}\right\}\right)\right]\int_{-\infty}^{0}\mathrm{d}v\,\exp\left[\frac{v}{\tau}\right]\delta(t'+s-u-v)
\end{aligned}
\qquad \text{(A2-11)}
$$

When applying the sifting property of the Dirac delta function, we note that the integral over $v$ is nonzero for $-\infty < t'+s-u \leq 0$, that is, for $t'+s \leq u$. Thus we have:

$$
\begin{aligned}
(2.iii) &= -\left(\frac{A}{\sqrt{2\pi}\sigma}\right)^3\int_{-\infty}^{\infty}\mathrm{d}t'\,\exp\left[-\frac{(t'-T_{ij})^2}{2\sigma^2}\right]\exp\left[-\frac{t'^2}{2\sigma^2}\right]\int_{0}^{\infty}\mathrm{d}s\int_{t'+s}^{\infty}\mathrm{d}u\,\exp\left[-\frac{(u-T_{ij})^2}{2\sigma^2}\right] \\
&\quad \exp\left[-\frac{1}{2\sigma^2}\left(s^2+2s\left\{t'+\frac{\sigma^2}{\tau}\right\}\right)\right]\exp\left[\frac{t'+s-u}{\tau}\right] \\
&= -\left(\frac{A}{\sqrt{2\pi}\sigma}\right)^3\exp\left[-\frac{T_{ij}^2}{\sigma^2}\right]\int_{-\infty}^{\infty}\mathrm{d}t'\,\exp\left[-\frac{1}{2\sigma^2}\left(2t'^2-2t'\left\{T_{ij}+\frac{\sigma^2}{\tau}\right\}\right)\right] \\
&\quad \int_{0}^{\infty}\mathrm{d}s\,\exp\left[-\frac{1}{2\sigma^2}\left(s^2+2t's\right)\right]\int_{t'+s}^{\infty}\mathrm{d}u\,\exp\left[-\frac{1}{2\sigma^2}\left(u^2-2u\left\{T_{ij}-\frac{\sigma^2}{\tau}\right\}\right)\right]
\end{aligned}
\qquad \text{(A2-12)}
$$

The integral over $u$ can be evaluated by using *Equation A2-6*, which yields:

$$
\begin{aligned}
(2.iii) &= -\frac{A^3}{4\pi\sigma^2}\exp\left[\frac{\sigma^2}{2\tau^2}-\frac{T_{ij}^2}{2\sigma^2}-\frac{T_{ij}}{\tau}\right]\int_{-\infty}^{\infty}\mathrm{d}t'\,\exp\left[-\frac{1}{2\sigma^2}\left(2t'^2-2t'\left\{T_{ij}+\frac{\sigma^2}{\tau}\right\}\right)\right] \\
&\quad \int_{0}^{\infty}\mathrm{d}s\,\exp\left[-\frac{1}{2\sigma^2}\left(s^2+2t's\right)\right]\left[1-\mathrm{erf}\left(\frac{1}{\sqrt{2}\sigma}\left\{t'+s-T_{ij}+\frac{\sigma^2}{\tau}\right\}\right)\right]
\end{aligned}
\qquad \text{(A2-13)}
$$

The second part of the integral over $s$ (involving the error function) will be solved numerically. For this purpose, we define $D_2'$ as:

$$
\begin{aligned}
D_2' &:= \frac{1}{\pi\sigma^2}\int_{-\infty}^{\infty}\mathrm{d}t'\,\exp\left[-\frac{1}{2\sigma^2}\left(2t'^2-2t'\left\{T_{ij}+\frac{\sigma^2}{\tau}\right\}\right)\right] \\
&\quad \int_{0}^{\infty}\mathrm{d}s\,\exp\left[-\frac{1}{2\sigma^2}\left(s^2+2t's\right)\right]\mathrm{erf}\left(\frac{1}{\sqrt{2}\sigma}\left\{t'+s-T_{ij}+\frac{\sigma^2}{\tau}\right\}\right)
\end{aligned}
\qquad \text{(A2-14)}
$$

For the first part of the integral over $s$ in *Equation A2-13*, we again use *Equation A2-6*, which results in:

$$(2.iii) = -\frac{A^3}{4\sqrt{2\pi}\sigma}\exp\left[\frac{\sigma^2}{2\tau^2} - \frac{T_{ij}^2}{2\sigma^2} - \frac{T_{ij}}{\tau}\right]\int_{-\infty}^{\infty}dt'\,\exp\left[-\frac{1}{2\sigma^2}\left(t'^2 - 2t'\left\{T_{ij} + \frac{\sigma^2}{\tau}\right\}\right)\right]$$
$$\left[1 + \mathrm{erf}\left(-\frac{t'}{\sqrt{2}\sigma}\right)\right] + \frac{A^3}{4}\exp\left[\frac{\sigma^2}{2\tau^2} - \frac{T_{ij}^2}{2\sigma^2} - \frac{T_{ij}}{\tau}\right]D_2'$$

(A2-15)

The first part of the integral over $t$ can be solved by applying *Equation A2-6* in the limit of $a \to -\infty$. For the second part we use *Equation A2-10*.

$$(2.iii) = -\frac{A^3}{4}\exp\left[\frac{\sigma^2}{\tau^2}\right]\left[1 - \mathrm{erf}\left(\frac{1}{2\sigma}\left\{T_{ij} + \frac{\sigma^2}{\tau}\right\}\right)\right] + \frac{A^3}{4}\exp\left[\frac{\sigma^2}{2\tau^2} - \frac{T_{ij}^2}{2\sigma^2} - \frac{T_{ij}}{\tau}\right]D_2'$$

(A2-16)

We now observe that defining $D_2$ in the following way:

$$D_2 = \frac{1}{\pi\sigma^2}\int_{-\infty}^{\infty}dt'\,\exp\left[-\frac{1}{2\sigma^2}\left(t' - \left\{T_{ij} + \frac{\sigma^2}{\tau}\right\}\right)^2\right]$$
$$\int_0^{\infty}ds\,\exp\left[-\frac{1}{2\sigma^2}(s+t')^2\right]\mathrm{erf}\left[\frac{1}{\sqrt{2}\sigma}\left(t' + s - \left\{T_{ij} - \frac{\sigma^2}{\tau}\right\}\right)\right]$$

(A2-17)

allows us to write $(2.iii)$ as:

$$(2.iii) = -\frac{A^3}{4}\exp\left[\frac{\sigma^2}{\tau^2}\right]\left[1 - \mathrm{erf}\left(\frac{1}{2\sigma}\left\{T_{ij} + \frac{\sigma^2}{\tau}\right\}\right) - D_2\right]$$

(A2-18)

## Addends of the second moment

By similar logic, all four addends of *Equation A2-2* (with four parts each) can be obtained. We list the results here:

### First Addend

$$(1.i) = \frac{A^4}{4}\exp\left[2\left(\frac{\sigma^2}{\tau^2} + \frac{T_{ij}}{\tau}\right)\right]\left[1 - \mathrm{erf}\left(\frac{1}{2\sigma}\left\{T_{ij} + 2\frac{\sigma^2}{\tau}\right\}\right)\right]^2$$

(A2-19)

$$(2.i) = -\frac{A^4}{4}\exp\left[2\frac{\sigma^2}{\tau^2}\right]\left[1 - \mathrm{erf}\left(\frac{1}{2\sigma}\left\{T_{ij} + 2\frac{\sigma^2}{\tau}\right\}\right)\right]\left[1 + \mathrm{erf}\left(\frac{1}{2\sigma}\left\{T_{ij} - 2\frac{\sigma^2}{\tau}\right\}\right)\right]$$

(A2-20)

$$(3.i) = -\frac{A^4}{4}\exp\left[2\frac{\sigma^2}{\tau^2}\right]\left[1 - \mathrm{erf}\left(\frac{1}{2\sigma}\left\{T_{ij} + 2\frac{\sigma^2}{\tau}\right\}\right)\right]\left[1 + \mathrm{erf}\left(\frac{1}{2\sigma}\left\{T_{ij} - 2\frac{\sigma^2}{\tau}\right\}\right)\right]$$

(A2-21)

$$(4.i) = \frac{A^4}{4}\exp\left[2\left(\frac{\sigma^2}{\tau^2} - \frac{T_{ij}}{\tau}\right)\right]\left[1 + \mathrm{erf}\left(\frac{1}{2\sigma}\left\{T_{ij} - 2\frac{\sigma^2}{\tau}\right\}\right)\right]^2$$

(A2-22)

### Second Addend

$$(1.ii) = \frac{A^3}{4}\exp\left[3\frac{\sigma^2}{\tau^2} + 2\frac{T_{ij}}{\tau}\right]\left[1 - 2\mathrm{erf}\left(\frac{1}{2\sigma}\left\{T_{ij} + 3\frac{\sigma^2}{\tau}\right\}\right) + C_1\right]$$

(A2-23)

$$(2.ii) = -\frac{A^3}{4}\exp\left[\frac{\sigma^2}{\tau^2}\right]\left[1 - \mathrm{erf}\left(\frac{1}{2\sigma}\left\{T_{ij} + \frac{\sigma^2}{\tau}\right\}\right) + \mathrm{erf}\left(\frac{1}{2\sigma}\left\{T_{ij} - \frac{\sigma^2}{\tau}\right\}\right) - C_2\right]$$

(A2-24)

$$(3.ii) = -\frac{A^3}{4}\exp\left[\frac{\sigma^2}{\tau^2}\right]\left[1 - \mathrm{erf}\left(\frac{1}{2\sigma}\left\{T_{ij}+\frac{\sigma^2}{\tau}\right\}\right) + \mathrm{erf}\left(\frac{1}{2\sigma}\left\{T_{ij}-\frac{\sigma^2}{\tau}\right\}\right) - C_2\right] \tag{A2-25}$$

$$(4.ii) = \frac{A^3}{4}\exp\left[3\frac{\sigma^2}{\tau^2}-2\frac{T_{ij}}{\tau}\right]\left[1 + 2\mathrm{erf}\left(\frac{1}{2\sigma}\left\{T_{ij}-3\frac{\sigma^2}{\tau}\right\}\right) + C_4\right] \tag{A2-26}$$

with the integral terms:

$$C_1 = \frac{1}{\sqrt{2\pi}\sigma}\int dt'\, \exp\left[-\frac{1}{2\sigma^2}\left(t' - \left\{T_{ij}+2\frac{\sigma^2}{\tau}\right\}\right)^2\right]$$
$$\cdot\mathrm{erf}^2\left[\frac{1}{\sqrt{2}\sigma}\left\{t'+\frac{\sigma^2}{\tau}\right\}\right] \tag{A2-27}$$

$$C_2 = \frac{1}{\sqrt{2\pi}\sigma}\int dt'\, \exp\left[-\frac{1}{2\sigma^2}\left(t' - T_{ij}\right)^2\right]$$
$$\cdot\mathrm{erf}\left[\frac{1}{\sqrt{2}\sigma}\left\{t'+\frac{\sigma^2}{\tau}\right\}\right]\mathrm{erf}\left[\frac{1}{\sqrt{2}\sigma}\left\{t'-\frac{\sigma^2}{\tau}\right\}\right] \tag{A2-28}$$

$$C_4 = \frac{1}{\sqrt{2\pi}\sigma}\int dt'\, \exp\left[-\frac{1}{2\sigma^2}\left(t' - \left\{T_{ij}-2\frac{\sigma^2}{\tau}\right\}\right)^2\right]$$
$$\cdot\mathrm{erf}^2\left[\frac{1}{\sqrt{2}\sigma}\left\{t'-\frac{\sigma^2}{\tau}\right\}\right] \tag{A2-29}$$

**Third Addend**

$$(1.iii) = \frac{A^3}{4}\exp\left[3\frac{\sigma^2}{\tau^2}+2\frac{T_{ij}}{\tau}\right]\left[1 - \mathrm{erf}\left(\frac{1}{2\sigma}\left\{T_{ij}+3\frac{\sigma^2}{\tau}\right\}\right) + D_1\right] \tag{A2-30}$$

$$(2.iii) = -\frac{A^3}{4}\exp\left[\frac{\sigma^2}{\tau^2}\right]\left[1 - \mathrm{erf}\left(\frac{1}{2\sigma}\left\{T_{ij}+\frac{\sigma^2}{\tau}\right\}\right) - D_2\right] \tag{A2-31}$$

$$(3.iii) = -\frac{A^3}{4}\exp\left[\frac{\sigma^2}{\tau^2}\right]\left[1 + \mathrm{erf}\left(\frac{1}{2\sigma}\left\{T_{ij}-\frac{\sigma^2}{\tau}\right\}\right) + D_3\right] \tag{A2-32}$$

$$(4.iii) = \frac{A^3}{4}\exp\left[3\frac{\sigma^2}{\tau^2}-2\frac{T_{ij}}{\tau}\right]\left[1 + \mathrm{erf}\left(\frac{1}{2\sigma}\left\{T_{ij}-3\frac{\sigma^2}{\tau}\right\}\right) - D_4\right] \tag{A2-33}$$

with the integral terms:

$$D_1 = \frac{1}{\pi\sigma^2}\int dt'\, \exp\left[-\frac{1}{2\sigma^2}\left(t' - \left\{T_{ij}+\frac{\sigma^2}{\tau}\right\}\right)^2\right]$$
$$\cdot\int_0^\infty ds\, \exp\left[-\frac{1}{2\sigma^2}\left(s + \left\{t'+2\frac{\sigma^2}{\tau}\right\}\right)^2\right]$$
$$\cdot\mathrm{erf}\left[\frac{1}{\sqrt{2}\sigma}\left(t'+s - \left\{T_{ij}+\frac{\sigma^2}{\tau}\right\}\right)\right] \tag{A2-34}$$

$$D_2 = \frac{1}{\pi\sigma^2}\int dt'\, \exp\left[-\frac{1}{2\sigma^2}\left(t' - \left\{T_{ij}+\frac{\sigma^2}{\tau}\right\}\right)^2\right]$$
$$\cdot\int_0^\infty ds\, \exp\left[-\frac{1}{2\sigma^2}(s+t')^2\right]$$
$$\cdot\mathrm{erf}\left[\frac{1}{\sqrt{2}\sigma}\left(t'+s - \left\{T_{ij}-\frac{\sigma^2}{\tau}\right\}\right)\right] \tag{A2-35}$$

$$D_3 = \frac{1}{\pi\sigma^2} \int dt' \, \exp\left[-\frac{1}{2\sigma^2}\left(t' - \left\{T_{ij} - \frac{\sigma^2}{\tau}\right\}\right)^2\right]$$
$$\cdot \int_{-\infty}^{0} ds \, \exp\left[-\frac{1}{2\sigma^2}(s+t')^2\right]$$
$$\cdot \mathrm{erf}\left[\frac{1}{\sqrt{2}\sigma}\left(t' + s - \left\{T_{ij} + \frac{\sigma^2}{\tau}\right\}\right)\right] \tag{A2-36}$$

$$D_4 = \frac{1}{\pi\sigma^2} \int dt' \, \exp\left[-\frac{1}{2\sigma^2}\left(t' - \left\{T_{ij} - \frac{\sigma^2}{\tau}\right\}\right)^2\right]$$
$$\cdot \int_{-\infty}^{0} ds \, \exp\left[-\frac{1}{2\sigma^2}\left(s + \left\{t' - 2\frac{\sigma^2}{\tau}\right\}\right)^2\right]$$
$$\cdot \mathrm{erf}\left[\frac{1}{\sqrt{2}\sigma}\left(t' + s - \left\{T_{ij} - \frac{\sigma^2}{\tau}\right\}\right)\right] \tag{A2-37}$$

**Fourth Addend**

$$(1.iv) = \frac{A^2}{2}\exp\left[4\frac{\sigma^2}{\tau^2} + 2\frac{T_{ij}}{\tau}\right]\left[1 - \mathrm{erf}\left(\frac{1}{2\sigma}\left\{T_{ij} + 4\frac{\sigma^2}{\tau}\right\}\right)\right] \tag{A2-38}$$

$$(2.iv) = (3.iv) = 0 \tag{A2-39}$$

$$(4.iv) = \frac{A^2}{2}\exp\left[4\frac{\sigma^2}{\tau^2} - 2\frac{T_{ij}}{\tau}\right]\left[1 + \mathrm{erf}\left(\frac{1}{2\sigma}\left\{T_{ij} - 4\frac{\sigma^2}{\tau}\right\}\right)\right] \tag{A2-40}$$

By collecting all 16 terms, we will obtain the average squared weight change. To calculate the variance, we also need the squared average weight change, which we will calculate in the next section.

## Average weight change

The average weight change for odd learning windows is given by (cmp. *Equation 1* in the main text):

$$\langle \Delta w_{ij} \rangle = \int_{-\infty}^{\infty} dt\, W(t) C_{ij}(t)$$
$$= \mu \int_{0}^{\infty} dt\, \exp\left(-\frac{t}{\tau}\right) C_{ij}(t) - \mu \int_{-\infty}^{0} dt\, \exp\left(\frac{t}{\tau}\right) C_{ij}(t)$$
$$= \mu \int_{0}^{\infty} dt\, \exp\left(-\frac{t}{\tau}\right) \int_{-\infty}^{\infty} dt'\, f_i(t') f_j(t' + t)$$
$$- \mu \int_{-\infty}^{0} dt\, \exp\left(\frac{t}{\tau}\right) \int_{-\infty}^{\infty} dt'\, f_i(t') f_j(t' + t) \tag{A2-41}$$

We again neglect the theta modulation of the firing fields. Evaluating the first addend yields:

$$\mu \int_0^\infty \mathrm{d}t \exp\left(-\frac{t}{\tau}\right) \int_{-\infty}^\infty \mathrm{d}t' f_i(t') f_j(t'+t)$$

$$= \frac{A^2\mu}{2\pi\sigma^2} \int_0^\infty \mathrm{d}t \exp\left(-\frac{t}{\tau}\right) \int_{-\infty}^\infty \mathrm{d}t' \exp\left(-\frac{t'^2}{2\sigma^2}\right) \exp\left(-\frac{(t+t'-T_{ij})^2}{2\sigma^2}\right)$$

$$= \frac{A^2\mu}{2\pi\sigma^2} \int_0^\infty \mathrm{d}t \exp\left(-\frac{t}{\tau}\right) \int_{-\infty}^\infty \mathrm{d}t' \exp\left(-\frac{t^2 + 2t'^2 + T_{ij}^2 + 2tt' - 2tT_{ij} - 2t'T_{ij}}{2\sigma^2}\right)$$

$$= \frac{A^2\mu}{2\pi\sigma^2} \exp\left(-\frac{T_{ij}^2}{2\sigma^2}\right) \int_0^\infty \mathrm{d}t \exp\left(-\frac{t^2 - 2\left(T_{ij} - \frac{\sigma^2}{\tau}\right)t}{2\sigma^2}\right) \int_{-\infty}^\infty \mathrm{d}t' \exp\left(-\frac{t'^2 + (t - T_{ij})t'}{\sigma^2}\right)$$

$$= \frac{A^2\mu}{2\sqrt{\pi}\sigma} \exp\left(-\frac{T_{ij}^2}{2\sigma^2}\right) \int_0^\infty \mathrm{d}t \exp\left(-\frac{t^2 - 2\left(T_{ij} - \frac{\sigma^2}{\tau}\right)t}{2\sigma^2}\right) \exp\left(\frac{(t - T_{ij})^2}{4\sigma^2}\right) \quad \text{(A2-42)}$$

$$= \frac{A^2\mu}{2\sqrt{\pi}\sigma} \exp\left(-\frac{T_{ij}^2}{4\sigma^2}\right) \int_0^\infty \mathrm{d}t \exp\left(-\frac{t^2 - 2\left(T_{ij} - 2\frac{\sigma^2}{\tau}\right)t}{4\sigma^2}\right)$$

$$= \frac{A^2\mu}{2} \exp\left(-\frac{T_{ij}^2}{4\sigma^2}\right) \exp\left[\frac{\left(T_{ij} - 2\frac{\sigma^2}{\tau}\right)^2}{4\sigma^2}\right] \left[1 + \mathrm{erf}\left(\frac{T_{ij} - 2\frac{\sigma^2}{\tau}}{2\sigma}\right)\right]$$

$$= \frac{A^2\mu}{2} \exp\left(\frac{\sigma^2}{\tau^2} - \frac{T_{ij}}{\tau}\right) \left[1 + \mathrm{erf}\left(\frac{T_{ij} - 2\frac{\sigma^2}{\tau}}{2\sigma}\right)\right]$$

The second addend can be similarly evaluated:

$$-\mu \int_{-\infty}^0 \mathrm{d}t \exp\left(\frac{t}{\tau}\right) \int_{-\infty}^\infty \mathrm{d}t' f_i(t') f_j(t'+t)$$

$$= -\frac{A^2\mu}{2\sqrt{\pi}\sigma} \exp\left(-\frac{T_{ij}^2}{4\sigma^2}\right) \int_{-\infty}^0 \mathrm{d}t \exp\left(-\frac{t^2 - 2\left(T_{ij} + 2\frac{\sigma^2}{\tau}\right)t}{4\sigma^2}\right) \quad \text{(A2-43)}$$

$$= -\frac{A^2\mu}{2} \exp\left(-\frac{T_{ij}^2}{4\sigma^2}\right) \exp\left[\frac{\left(T_{ij} + 2\frac{\sigma^2}{\tau}\right)^2}{4\sigma^2}\right] \left[1 - \mathrm{erf}\left(\frac{T_{ij} + 2\frac{\sigma^2}{\tau}}{2\sigma}\right)\right]$$

$$= -\frac{A^2\mu}{2} \exp\left(\frac{\sigma^2}{\tau^2} + \frac{T_{ij}}{\tau}\right) \left[1 - \mathrm{erf}\left(\frac{T_{ij} + 2\frac{\sigma^2}{\tau}}{2\sigma}\right)\right]$$

The average weight change thus reads:

$$\frac{1}{\mu}\langle\Delta w_{ij}\rangle = \frac{A^2}{2}\left\{ \exp\left(\frac{\sigma^2}{\tau^2} - \frac{T_{ij}}{\tau}\right)\left[1 + \mathrm{erf}\left(\frac{T_{ij} - 2\frac{\sigma^2}{\tau}}{2\sigma}\right)\right] - \exp\left(\frac{\sigma^2}{\tau^2} + \frac{T_{ij}}{\tau}\right)\left[1 - \mathrm{erf}\left(\frac{T_{ij} + 2\frac{\sigma^2}{\tau}}{2\sigma}\right)\right]\right\} \quad \text{(A2-44)}$$

*Equation A2-44* might show numerical instabilities for small $\tau$. These instabilities can be fixed using the following approximation for the error function proposed by *Abramowitz, 1974*:

$$\mathrm{erf}(x) \approx 1 - (a_1 t + a_2 t^2 + a_3 t^3)\exp(-x^2), \quad t = \frac{1}{1 + px}, \quad x \geq 0, \quad \text{(A2-45)}$$

where $p = 0.47047$, $a_1 = 0.3480242$, $a_2 = -0.0958798$, $a_3 = 0.7478556$. Along with a new set of variables

$$x_1 = \frac{1}{2\sigma}\left(2\frac{\sigma^2}{\tau} - T_{ij}\right) \qquad x_2 = \frac{1}{2\sigma}\left(2\frac{\sigma^2}{\tau} + T_{ij}\right) \geq 0 \text{ because } T_{ij} \geq 0$$

$$u_1 = \frac{1}{1 + px_1} \qquad u_1' = \frac{1}{1 - px_1}, \qquad\qquad u_2 = \frac{1}{1 + px_2},$$

the approximation yields

$$\frac{1}{\mu}\langle\Delta w_{ij}\rangle = \frac{A^2}{2}\begin{cases} \exp\left(-\frac{T_{ij}^2}{4\sigma^2}\right)\left[a_1 u_1 + a_2 u_1^2 + a_3 u_1^3 - a_1 u_2 - a_2 u_2^2 - a_3 u_2^3\right], & x_1 \geq 0, x_2 \geq 0 \\ 2\exp\left(\frac{\sigma^2}{\tau^2} - \frac{T_{ij}}{\tau}\right) - \exp\left(-\frac{T_{ij}^2}{4\sigma^2}\right)\left[a_1(u_1' + u_2) + a_2(u_1'^2 + u_2^2) + a_3(u_1'^3 + u_2^3)\right], & x_1 < 0 \end{cases} \tag{A2-46}$$

Note that the exponential with the $\frac{\sigma^2}{\tau^2}$ term vanishes when $x_1 \geq 0$, and only one addend contains the term for $x_1 < 0$. Therefore, this approximation results in improved numerical stability for small $\tau$. *Equation A2-46* is shown in *Figure 4A*.

## Variance and signal-to-noise ratio of the weight change

With all of the above results, we are now ready to state the variance and signal-to-noise ratio of the weight change:

$$\begin{aligned} \frac{1}{\mu^2}\mathrm{var}(\Delta w_{ij}) =& \frac{1}{\mu^2}\left(\langle\Delta w_{ij}^2\rangle - \langle\Delta w_{ij}\rangle^2\right) \\ =& \frac{A^3}{4}\left\{\exp\left[3\frac{\sigma^2}{\tau^2} + 2\frac{T_{ij}}{\tau}\right]\left[2 + C_1 + D_1 - 3\mathrm{erf}\left(\frac{T_{ij} + 3\frac{\sigma^2}{\tau}}{2\sigma}\right)\right]\right. \\ &- \exp\left[\frac{\sigma^2}{\tau^2}\right]\left[4 - 2C_2 - D_2 + D_3 - 3\mathrm{erf}\left(\frac{T_{ij} + \frac{\sigma^2}{\tau}}{2\sigma}\right) + 3\mathrm{erf}\left(\frac{T_{ij} - \frac{\sigma^2}{\tau}}{2\sigma}\right)\right] \\ &\left.+ \exp\left[3\frac{\sigma^2}{\tau^2} - 2\frac{T_{ij}}{\tau}\right]\left[2 + C_4 - D_4 + 3\mathrm{erf}\left(\frac{T_{ij} - 3\frac{\sigma^2}{\tau}}{2\sigma}\right)\right]\right\} \\ &+ \frac{A^2}{2}\left\{\exp\left[4\frac{\sigma^2}{\tau^2} + 2\frac{T_{ij}}{\tau}\right]\left[1 - \mathrm{erf}\left(\frac{1}{2\sigma}\left\{T_{ij} + 4\frac{\sigma^2}{\tau}\right\}\right)\right]\right. \\ &\left.+ \exp\left[4\frac{\sigma^2}{\tau^2} - 2\frac{T_{ij}}{\tau}\right]\left[1 + \mathrm{erf}\left(\frac{1}{2\sigma}\left\{T_{ij} - 4\frac{\sigma^2}{\tau}\right\}\right)\right]\right\} \end{aligned} \tag{A2-47}$$

The signal-to-noise ratio is then given by:

$$\mathrm{SNR} = \frac{\langle\Delta w_{ij}\rangle}{\mathrm{std}(\Delta w_{ij})} = \frac{\langle\Delta w_{ij}\rangle}{\sqrt{\mathrm{var}(\Delta w_{ij})}} \tag{A2-48}$$

*Equation A2-48* (with the variance from *Equation A2-47* and the mean from *Equation A2-44*) is shown in *Figure 4C*. We observe that the analytical solution fits the numerical solution well for $\tau \geq 0.1$ s but numerical instabilities cause it to diverge for $\tau \leq 0.1$ s.

The numerical instability for $\tau \leq 0.1$ s is likely due to a combination of two factors: the exponential terms $\exp[\sigma^2/\tau^2]$ become very large for small tau, and large arguments in the error function cause the terms $(1 \pm \mathrm{erf}(.))$ to be very close to zero. The product of the two is numerically unstable for small tau. Unfortunately, unlike in the case of the average weight change, we did not find an approximation which canceled out these exponential terms in the noise.

