## [Decision Letter]

**Acceptance summary:**

One of the major challenges of cortical circuits is to learn associations between events that are separated by long time periods, given that spike-timing-dependent plasticity operates on short time scales. In the Hippocampus, a structure critical for memory formation, phase precession is known to compress the sequential activation of place fields to the theta-cycle (~8Hz) time period. Reifenstein et al., describe a simple yet elegant mathematical principle through which theta phase precession contributes to learning the sequential order by which place fields are activated.

**Decision letter after peer review:**

[Editors’ note: the authors submitted for reconsideration following the decision after peer review. What follows is the decision letter after the first round of review.]

Thank you for submitting your work entitled "Synaptic learning rules for sequence learning" for consideration by *eLife*. Your article has been reviewed by 4 peer reviewers, including Martin Vinck as the Reviewing Editor and Reviewer #1, and the evaluation has been overseen by a Senior Editor. The following individuals involved in review of your submission have agreed to reveal their identity: Francesco P Battaglia (Reviewer #2); Frances Chance (Reviewer #4).

Our decision has been reached after consultation between the reviewers. Based on these discussions and the individual reviews below, we regret to inform you that your work will not be considered further for publication in *eLife*. However, *eLife* would welcome a substantially improved manuscript that addresses concerns raised; this would be treated as a new submission, but likely go to the same reviewers.

The reviewers acknowledged that the study addresses an important topic. They also applauded the rigor and elegance of the analytical approach. However, reviewers individually, and in subsequent discussion, expressed the concern that the physiological relevance of the findings is far from clear; this point would require a substantial amount of new simulations and models. They furthermore commented that the generation and storage of sequences remains unclear, again requiring substantial additions to the manuscript. Reviewers therefore recommended that, at present, the manuscript appears to be more suited for a more specialized journal.

*Reviewer #1:*

This paper develops a model of the way in which phase precession modulates synaptic plasticity. The idea and derivations are simple and easy to follow. The results, while not surprising, are overall interesting and important for researchers on phase precession and sequence learning. There are some useful analytical approximations in the paper. I have several comments:

1. The paper is all based on pairwise STDP.

How robust are these results when we consider perhaps more realistic STDP rules like triplet STDP? Perhaps this is something to discuss or explore, because it is not a priori obvious to me.

2. What are the widths reported in the literature for hippocampus? With all the recent literature on the dependence of STDP in vitro on Ca^2+^ levels, one has to take this with a grain of salt of course. I would think it's around 100ms which would make the benefit small?

3. The approximation of theta as an oscillator that shows no dampening is of course not realistic; in reality autocorrelation functions will show decreasing sidelobes. It's maybe not a problem, but could actually benefit your model.

4. To say that phase precession benefits sequence learning is maybe not the whole story. It seems that in general long STDP kernels benefit sequence learning for place fields, and they do this equally well for phase or no phase precession. If the STDP kernels are short, sequence learning is more difficult (and requires huge place field overlap), and phase precession is beneficial for that.

How does benefit interact with place field overlap? If the place fields are highly overlapping, then how does STDP kernel size regulate the sequence learning? Are longer STDP kernels invariantly better for sequence learning in the hippocampus? Or does this depend on place field separation. In other words are there are some scenarios where short STDP kernels have a clear benefit and where phase precession then gives a huge boost?

*Reviewer #2:*

Reifenstein and Kempter propose an analytical formulation for synaptic plasticity dynamics with STDP and phase precession as observed in hippocampal place cells.

The main result is that phase precession increases the slope of the 2-cell cross-correlation around the origin, which is the key driver of plasticity under asymmetric STDP, therefore improving the encoding of sequences in the synaptic matrix.

While the overall concept of phase precession favoring time compression of sequences and plasticity (when combined with STDP) has been present in the literature since the seminal Skaggs and McNaughton, 1996 paper, the novel contribution of this study is the elegant analytical formulation of the effect, which can be very useful to embed this effect into a network model. As a suggestion of a further direction, one could look at models (e.g. Tsodyks et al., Hippocampus, 1996) where asymmetries in synaptic connections are driver for phase precession. One could use this formulation for e.g. seeing how experience may induce phase precessing place field by shaping synaptic connections (maybe starting from a small symmetry breaking term in the initial condition).

The analytical calculation seems crystal clear to me (and quite simple, once one finds the right framework)

*Reviewer #3:*

The study uses analytical and numerical approaches to quantify conditions in which spike timing-dependent plasticity (STDP) and theta phase precession may promote sequence learning. The strengths of the study are that the question is of general interest and the analytical approach, in so far as it can be applied, is quite rigorous. The weaknesses are that the extent to which the conclusions would hold in more physiological scenarios is not considered, and that the study does not investigate sequences but rather the strength of synaptic connections between sequentially activated neurons.

1. While the stated focus in on sequences, the key results are based on measures of synaptic weight between sequentially activated neurons. Given the claims of the study, a more relevant readout might be generation of sequences by the trained network.

2. The target network appears very simple. Assuming it can generate sequences, it's unclear whether the training rule would function under physiologically relevant conditions. For example, can the network trained in this way store multiple sequences? To what extent do sequences interfere with one another?

3. In a behaving animal movement speed varies considerably, with the consequence that the time taken to cross a place field may vary by an order of magnitude. I think it's important to consider the implications that this might have for the results.

4. Phase precession, STDP and sequence learning have been considered in previous work (e.g. Sato and Yamaguchi, Neural Computation, 2003; Shen et al., Advances in Cognitive Neurodynamics ICCN, 2007; Masquelier et al., J. Neurosci. 2009; Chadwick et al., *eLife* 2016). These previous approaches differ to various degrees from the present work, but each offers alternative suggestions for how STDP and phase precession could interact during memory. It's not clear what the advantages are of the framework proposed here.

5. While theta sequences are the focus of the introduction, many of the same arguments could be applied to compressed representations during sharp wave ripple events. This may be worth considering. Also, given the model involves excitatory connections between neurons that represent sequences, the relevance here may be more to CA3 were such connectivity is more common, rather than CA1 which is the focus of many of the studies cited in the introduction.

*Reviewer #4:*

This manuscript argues that phase precession enhances the learning of sequence learning by compressing a slower behavioral sequence, for example movement of an animal through a sequence of place fields, into a faster time scales associated with synaptic plasticity. The authors examine the synaptic weight change between pairs of neurons encoding different events in the behavioral sequence and find that phase precession enhances sequence learning when the learning rule is asymmetric over a relatively narrow time window (assuming the behavioral events encoded by the two neurons overlap, ie the place fields of the neurons overlap). For wider time windows, however, phase precession does not appear to convey any advantage.

I thought the study was interesting – the idea that phase precession "compresses" sequences into theta cycles has been around for a bit, but this is the first study that I've seen that does analysis at this level. I think many researchers who are interested in temporal coding would find the work very interesting.

I did, however, have a little trouble understanding what conclusions the study draws about the brain (if we are supposed to draw any). The authors conclude that phase precession facilitates if the learning window is shorter than a theta cycle – that seems in line with published STDPs rules from slice studies. However, Figure 4 seems to imply that the authors have recovered a 1 second learning window from Bittner's data – are they suggesting that phase precession is not an asset for the learning in that study (or did I miss something)? Are there predictions to be made about how place fields must be spaced for optimal sequence learning?

Also, I'd be curious to know how the authors analysis fits in with replay – is the assumption that neuromodulation is changing the time window or other learning dynamics?

---

## [Author Response]

[Editors’ note: the authors resubmitted a revised version of the paper for consideration. What follows is the authors’ response to the first round of review.]

Reviewer #1:This paper develops a model of the way in which phase precession modulates synaptic plasticity. The idea and derivations are simple and easy to follow. The results, while not surprising, are overall interesting and important for researchers on phase precession and sequence learning. There are some useful analytical approximations in the paper. I have several comments:1. The paper is all based on pairwise STDP.How robust are these results when we consider perhaps more realistic STDP rules like triplet STDP? Perhaps this is something to discuss or explore, because it is not a priori obvious to me.

In “pairwise STDP”, pairs of presynaptic and postsynaptic spikes are considered. Conversely, “triplet STDP” considers triplet motifs of spiking (either 2 presynaptic – 1 postsynaptic or 2 postsynaptic – 1 presynaptic). Triplet STDP models allow to account for a number of experimental findings that pairwise STDP fails to reproduce, for example the dependence on the repetition frequency of spike pairs. However, it is unclear whether our results on sequence learning still hold for generic triplet STDP rules.

To investigate the relative weight change (forward weight minus backward weight), we reproduced results like the ones shown in Figure 3 of our manuscript for generic versions of the triplet rule from Pfister and Gerstner, (2006), who fitted triplet rule models to data from the hippocampus. Their model consists of four terms: pairwise potentiation, pairwise depression, triplet potentiation, and triplet depression. To be able to compare triplet STDP models with pairwise STDP models, we first simulated pairwise potentiation and pairwise depression according to the learning rule from Bi and Poo, (1998). The results resembled very much our Figure 3 (see Author response image 1) because the Bi-and-Poo rule is close to the perfectly odd learning rule used for Figures 3 and 4 (see also the new simulations results shown in Figure 4C, for example for the Bi-and-Poo rule). We then added triplet terms with the parameters of the minimal model described in Table 4 (“All-to-All”, “Minimal” model, Pfister and Gerstner, 2006). This “minimal” model, which included only one triplet term to pairwise STDP (the “triplet potentiation term”, i.e., a 1-pre-2-post term), was regarded as the best model in terms of number of free parameters and fitting error. We found that the results were very similar to the pairwise Bi-and-Poo rule (see Author response image 1).

The small difference between pairwise and triplet STDP is probably due to the fact that the time constant for the triplet potentiation term is only 40 ms, which is shorter than the average ISI in our simulations with values typically > 50 ms (minimum average ISIs is 50 ms in the center of the firing field with a peak rate of 20 spikes/s; see, e.g., Figure 2A). This comparison therefore suggests that we can neglect triplet potentiation in our framework because the time constant of the triplet term is low enough.

**Author response image 1. respfig1:** Comparison of pairwise and triplet STDP. (**A**) Average weight change for the pairwise Bi-and-Poo learning rule (circles) and the minimal triplet model from Pfister and Gerstner, 2006 (squares). Bluish symbols represent phase precession, reddish symbols represent phase locking. Note that we have normalized the weight changes here (by the peak of the respective phase-locking curve) because the parameters *A^+^* and *A^-^* differ in Bi and Poo (1998) as compared to Pfister and Gerstner, (2006) as they were fitted to different data sets. In Figure 3A in the manuscript we show the un-normalized weight change because we choose the learning-rate parameter 𝜇=1 for mathematical convenience. (**B**) Benefit of phase precession as defined in the manuscript (equation 6) for the Bi-and-Poo learning rule (circles) and the minimal triplet model (squares). (**C**) Signal-to-noise ratio for Bi-and-Poo learning rule and the minimal triplet model. Symbols and colors as in (A). We note that the results in B and C are independent of the value of the learning-rate parameter.

We note, however, that we found larger differences (for weight changes, benefit, and SNR) when we used (instead of the “minimal” model) the “All-to-All”, “Full” model from Table 4 in Pfister and Gerstner (2006), which included triplet depression, i.e., a 2-pre-1-post term. This marked difference between “minimal” and “full” models in our simulations was surprising because Pfister and Gerstner, (2006) observed very similar fitting errors. A closer inspection revealed the origin of this difference: the triplet depression term has a time constant τ _x_ = 946 ms, which leads in our simulations to a strong accumulation of the corresponding dynamic variable *r2* that keeps track of presynaptic events (equation 2 in Pfister and Gerstner, 2006). This accumulation is particularly relevant in our simulations in which we consider widths of firing fields and time lags between firing fields on the order of one second. On the other hand, the data to which Pfister and Gerstner (2006) fitted their model did not critically rely on such long delays; instead the data were dominated by pairs and triples with time differences on a 10 ms scale. Therefore, the long-time constant of τ _x_ = 946 ms of triplet depression should not be a critical parameter of their model. This fact was recognized by Pfister and Gerstner, (2006), who showed that “minimal” triplet learning rules are almost as good as the “full” ones but have two parameters less. Therefore the “minimal” model was regarded as the best model. Because this “minimal” model did not change the outcome in our sequence learning paradigm, we conclude that our results obtained for pairwise STDP are robust with respect to effects originating from triplets of spikes.

To indicate in the manuscript that our results are robust when triplet STDP is considered, we have added simulation results of the “minimal” triplet rule from Pfister and Gerstner, (2006) in Figure 4C and have included a paragraph on triplet rules in the Discussion.

2. What are the widths reported in the literature for hippocampus? With all the recent literature on the dependence of STDP in vitro on Ca^2+^ levels, one has to take this with a grain of salt of course. I would think it's around 100ms which would make the benefit small?

The reported time constants in the literature for hippocampus are on the order of 15-30 ms (e.g. Abbott and Nelson, 2000; Bi and Poo, 2001; Wittenberg and Wang, 2006; Inglebert et al., 2020). In this case, the benefit is large (Figure 4B), and the “width” of a learning window, i.e., the range of the time interval in which weights are affected, appears to be in the range of 100 ms (see e.g. Figure 1 in Bi and Poo, 2001).

We agree with the reviewer that STDP depends on the Ca^2+^ level, as recent studies show (e.g. Inglebert et al., 2020). However, the total width of the STDP kernels rarely exceeds 100 ms — corresponding to ~50 ms for each lobe, positive and negative time lags. These widths are in line with the reported time constants of 15-30 ms mentioned above.

To create a stronger connection to the STDP literature, we added and discussed the following references to the manuscript: Froemke et al., 2005; Wittenberg and Wang, 2006; Inglebert et al., 2020. Furthermore, we estimated the SNR of the synaptic weight change for a number of experimental STDP kernels and included the results in Figure 4C, as a comparison to our theoretical results.

3. The approximation of theta as an oscillator that shows no dampening is of course not realistic; in reality autocorrelation functions will show decreasing sidelobes. It's maybe not a problem, but could actually benefit your model.

The reviewer is right in that we do not explicitly model dampening sidelobes in the spiking autocorrelation function. For narrow STDP windows (*K*≪ 1/ω ≪ σ ), however, dampening sidelobes would have no effect on the synaptic weight change because only the slope of the cross-correlation function around zero-time lag matters (Figure 2C,D and Equation 3 in the manuscript). Also for wide STDP windows (*K*≫ σ ), dampening sidelobes of the theta modulation would not cause a difference because we show (e.g. in Figure 4 for τ > 0.3 s) that “phase precession” and “phase locking” are basically identical to the case of no theta. In the Discussion (section “Key features of phase precession for temporal order-learning: generalization to non-periodic modulation of activity”) we even mention a scenario in which temporal-order learning could benefit from spike statistics similar to phase precession but in the absence of any periodic modulation.

Taken together, we think that the shape of the theta modulation is not a critical model assumption. To better emphasize this, we have added a brief note on irrelevant features of the autocorrelation already at the end of the paragraph following Equation 3.

4. To say that phase precession benefits sequence learning is maybe not the whole story. It seems that in general long STDP kernels benefit sequence learning for place fields, and they do this equally well for phase or no phase precession. If the STDP kernels are short, sequence learning is more difficult (and requires huge place field overlap), and phase precession is beneficial for that.

Indeed, phase precession can benefit temporal-order learning for short STDP kernels and overlapping firing fields, and this is one of our main findings. To make sure that this gets not (erroneously) generalized to wide STDP kernels, we checked our wording throughout the manuscript to be clear about the “short STDP kernels” condition. As a result, we added the words “for short synaptic learning windows” to the abstract.

On the other hand, wide STDP kernels also can facilitate temporal-order learning, but only if they are sufficiently asymmetric. Any symmetric component of the STDP kernel disturbs temporal-order learning, as we exemplify in Figure 4C by applying the learning window from Bittner et al., (2017) to our framework. We further fully agree with the reviewer that the temporal fine structure of spiking (phase precession vs. phase locking) becomes irrelevant for wide STDP kernels. For short kernels, however, phase precession makes all the difference (Figures 3 and 4).

To further clarify what we mean by “sequence learning”, we defined the terms “sequence learning” (in the Results, below the first equation) and “temporal-order learning” (in the Introduction) and replaced the more general term “sequence learning” by “temporal-order learning” at many suitable places in the manuscript.

How does benefit interact with place field overlap?

The larger the overlap, the stronger the benefit. This is shown in Figure 3B and described analytically in Equation 7 where we relate the benefit to the field separation *Tij* (which is the inverse of the overlap). We improved the text below Equation 7 to more strongly emphasize the relationship between the field separation *Tij* and the field overlap. We additionally made sure to include the field overlap in the summary sentence of the paragraph describing Figure 3B.

If the place fields are highly overlapping, then how does STDP kernel size regulate the sequence learning?

For overlapping fields (*Tij* = 𝜎), we show in Figure 4 that weight change (Figure 4A) and SNR (Figure 4C) increase with increasing STDP kernel width. We note that these results crucially depend on the STDP kernel being perfectly asymmetric. For learning windows with a symmetric component (dots in Figure 4C) the SNR is much lower and decreases for large widths. To better emphasize this in the manuscript, we included in Figure 4C further experimentally observed learning windows (for more details, see also our response to the next question by the reviewer).

Are longer STDP kernels invariantly better for sequence learning in the hippocampus?

Long STDP kernels are *not* invariantly better for temporal-order learning because it depends on the symmetry of the STDP kernel. A symmetric component of the STDP kernel reduces temporal-order learning — as we exemplify in Figure 4C by applying the very long (in the order of one second) and mostly symmetric learning window from Bittner et al. (2017) to our framework. Additionally, in the updated version of Figure 4C, we added several experimentally found learning windows (which also had symmetric components) from other studies (dots). SNRs were again below the blue line, which indicates the maximal SNR for purely asymmetric windows and phase precession. The most extreme case, i.e., a long (τ ≫ σ ) and purely asymmetric STDP kernels would yield the largest weight change and largest SNR but, in this scenario, phase precession would not alter the result compared to phase locking or no theta (as we show in Figure 4 A,B,C). However, long and predominantly asymmetric STDP windows have — to date and to the best of our knowledge — not been experimentally observed.

Or does this depend on place field separation.

For longer, asymmetric STDP kernels, the dependence of the weight change on place-field separation is given by Equation 9 and illustrated in Figure 5D: with increasing *T**ij*, the weight change quickly increases, reaches a maximum, and then slowly decreases. Figure 5E shows that the SNR increases with increasing *T* *ij*, but quickly settles on a constant value. We again note that weight change and SNR would be lower for learning windows with a symmetric component.

In other words are there are some scenarios where short STDP kernels have a clear benefit and where phase precession then gives a huge boost?

We assume that the reviewer means “benefit” as we use the term in the manuscript, i.e., the benefit of phase precession over phase locking (instead of short vs. long STDP kernels, which we discussed above). In that case, Figure 4B shows that short (τ < 100 ms) STDP kernels generate a clear benefit, i.e., phase precession leads to clearly larger weight changes than phase locking (Figure 4A); this result is also supported by Equation 7 (solid black line in Figure 4B), which shows how the benefit depends on all parameters of the model. The benefit is the stronger the smaller τ is. This effect is confirmed by the signal-to-noise ratio analysis in Figure 4C.

We believe that the last set of reviewer’s questions suggests that we should improve the manuscript to clarify the distinction between symmetric/asymmetric (mathematically even/odd) STDP kernels in the manuscript. We did so in the Results, and we also used more informative headlines for the subsections in the results.

Reviewer #2:Reifenstein and Kempter propose an analytical formulation for synaptic plasticity dynamics with STDP and phase precession as observed in hippocampal place cells.The main result is that phase precession increases the slope of the 2-cell cross-correlation around the origin, which is the key driver of plasticity under asymmetric STDP, therefore improving the encoding of sequences in the synaptic matrix.While the overall concept of phase precession favoring time compression of sequences and plasticity (when combined with STDP) has been present in the literature since the seminal Skaggs and McNaughton 1996 paper, the novel contribution of this study is the elegant analytical formulation of the effect, which can be very useful to embed this effect into a network model.

We thank the reviewer for this very positive assessment of our work and particularly for pointing out the appeal of the analytical approach.

As a suggestion of a further direction, one could look at models (eg Tsodyks et al., Hippocampus 1996) where asymmetries in synaptic connections are driver for phase precession. One could use this formulation for eg seeing how experience may induce phase precessing place field by shaping synaptic connections (maybe starting from a small symmetry breaking term in the initial condition).

We thank the reviewer for this interesting direction to extend our work. In the current manuscript, we assume that asymmetries in synaptic connections do not generate phase precession (in contrast to Tsodyks et al., 1996). We even assume, for simplicity of the analytical treatment, that recurrent connections do not affect the dynamics. We thus hypothesize that phase precession is not generated by the local, recurrent network; instead, phase precession is inherited or generated locally by a cellular/synaptic mechanism. After experience, the resulting asymmetric connections could indeed also generate phase precession (as demonstrated by the simulations by Tsodyks et al., 1996), and this phase precession could then even be similar to the one that has initially helped to shape synaptic connections. Finally, inherited or local cellularly/synaptically-generated phase precession and locally network-generated phase precession could interact (as reviewed, for example in Jaramillo and Kempter, 2017). We added this important line of thought to the Discussion (new section “Model assumptions”) of our manuscript.

The analytical calculation seems crystal clear to me (and quite simple, once one finds the right framework)

We thank the reviewer for this encouraging comment.

Reviewer #3:The study uses analytical and numerical approaches to quantify conditions in which spike timing-dependent plasticity (STDP) and theta phase precession may promote sequence learning. The strengths of the study are that the question is of general interest and the analytical approach, in so far as it can be applied, is quite rigorous.

We thank the reviewer for these positive comments on general interest and thorough analysis.

The weaknesses are that the extent to which the conclusions would hold in more physiological scenarios is not considered, and that the study does not investigate sequences but rather the strength of synaptic connections between sequentially activated neurons.

We regret that it became not clear enough how the conclusions of this theoretical work could be applied to more physiological scenarios, and why the predicted changes of synapses have strong implications on the replay of sequences. We try to respond to this general critique in detail below (see points 1-5).

1. While the stated focus in on sequences, the key results are based on measures of synaptic weight between sequentially activated neurons. Given the claims of the study, a more relevant readout might be generation of sequences by the trained network.

We agree that an appropriate readout of the result of learning would be the generation of sequences by a trained network. However, the fundamental basis of replay is a specific connectivity whereas the detailed characteristics of replay also depend on a variety of other parameters that define the neurons and the network. To illustrate the enormous complexity of the relation between the weights in a very simple network and the properties of replay, we refer the reviewer, for example, to Cheng, (2013) or Chenkov et al., (2017). Also the reviewer’s suggestions below (Sato and Yamaguchi, 2003; Shen et al., 2007) offer insights into the challenges of network simulations that include replay. To repeat such network simulations would be way beyond the scope of our manuscript, which tries to reveal the intricate relation between plasticity and weight changes. Because replay is indeed an important readout, we nevertheless thoroughly linked our work to the literature on the generation of sequences in trained networks; for example, we now also discuss the work by Gauy et al., (2018), Malerba and Bazhenov, (2019), and Gillett et al., (2020). Additionally, we have extensively discussed models of replay in the Discussion.

To better state our aims, to weaken our claims, and to scale down (possibly wrong) expectations, we added at end of the first paragraph of the Results (where we first mention “replay”) a remark on the scope of this work: “We note, however, that in what follows we do not simulate such a replay of sequences, which would depend also on a vast number of parameters that define the network; instead, we focus on the underlying changes in connectivity, which is the very basis of replay, and draw connections to replay in the Discussion.”

On the other hand, we note that we have removed the paragraph on replay speed because we felt that the numbers used for its estimation were questionable: (i) the delay of 10ms for the propagation of activity from one assembly to the next in Chenkov et al., (2017) might depend on the specific choice of parameters and (ii) the estimated spatial width of a place field (here 0.09m) is realistic but arbitrary. Much larger place fields exist. Therefore, the two numbers that are the basis for the estimation of the replay speed are variable and the replay speed (ratio of the two) might vary strongly.

2. The target network appears very simple. Assuming it can generate sequences, it's unclear whether the training rule would function under physiologically relevant conditions. For example, can the network trained in this way store multiple sequences? To what extent do sequences interfere with one another?

We agree with the reviewer that the anatomy of the target network appears very simple. However, the problem of evaluating weight changes in dependence upon phase precession, place field properties, and STDP parameters is quite complex. Though it is unclear whether physiologically relevant conditions can ever be achieved in a computational model (e.g., Almog and Korngren, 2016, J Neurophysiol), our model attempts to carefully reflect the essence of biological reality, and thus we consider our parameter settings as physiologically realistic as possible. Other theoretical work has demonstrated that multiple sequences can be stored in a network and that the memory capacity for sequences can be large (e.g. Leibold and Kempter, 2006; Trengove et al., 2013; Chenkov et al., 2017). We note that the corresponding network simulations are usually quite involved and typically depend on a vast number of parameters. We thus think that these kinds of network simulations are well beyond the scope of the current study.

We nevertheless address the topics of replay in general and multiple (and possibly overlapping) sequences in particular in the Discussion (sections “Other mechanisms and models for sequence learning” and “Replay of sequences and storage or multiple and overlapping sequences”), and now have added a note on storing multiple sequences and memory capacity.

3. In a behaving animal movement speed varies considerably, with the consequence that the time taken to cross a place field may vary by an order of magnitude. I think it's important to consider the implications that this might have for the results.

Running speed indeed affects field size and field distance (when measured in units of time). Because our theory investigates plasticity in dependence upon field size (which we quantify in units of time) and field separation (also in units of time, Figures 3 and 5; equations 4, 5 and 7; as well as the derivation of the maximal benefit in the paragraph after equation 7), our results include variations in running speed.

To make all this more explicit, we clarified in the manuscript (in the second paragraph after Equation 3, starting with “As a generic example.…”) that field width and field separation are measured in units of time (and not in units of length).

4. Phase precession, STDP and sequence learning have been considered in previous work (e.g. Sato and Yamaguchi, Neural Computation, 2003; Shen et al., Advances in Cognitive Neurodynamics ICCN, 2007; Masquelier et al., J. Neurosci. 2009; Chadwick et al., eLife 2016). These previous approaches differ to various degrees from the present work, but each offers alternative suggestions for how STDP and phase precession could interact during memory. It's not clear what the advantages are of the framework proposed here.

We thank the reviewer for the hint to these references and have included all of them in our manuscript. In particular, Sato and Yamaguchi, (2003) add a valuable contribution, investigating phase precession and STDP in a network of coupled phase-oscillators — a clear and rewarding approach, yet somewhat detached from biology.

We would like to point out that our formulation of phase precession and STDP intends to reflect the biological reality as close as possible. All individual components like phase precession, STDP, and place fields have been experimentally described. This is in contrast, to, e.g., phase precession in interneurons, as assumed by Chadwick et al., 2016. Compared to the cited new references, a major advantage of our approach is the analytical tractability. Our mathematical treatment of the problem yields a clear description of parameter dependencies — in contrast to, e.g., Shen et al., 2007 who investigate only one example of a small-network simulation and thus cannot predict the dependence of learning upon the various parameters.

Finally, Masquelier et al., (2009) offer an alternative approach to learning using phase coding and STDP, and Chadwick et al., (2016) nicely explain generation of phase precession via recurrent networks.

5. While theta sequences are the focus of the introduction, many of the same arguments could be applied to compressed representations during sharp wave ripple events. This may be worth considering. Also, given the model involves excitatory connections between neurons that represent sequences, the relevance here may be more to CA3 were such connectivity is more common, rather than CA1 which is the focus of many of the studies cited in the introduction.

The place field width is also in seconds, we now point that out clearer (in the second paragraph after Equation 3, starting with “As a generic example.…”).

Reviewer #4:This manuscript argues that phase precession enhances the learning of sequence learning by compressing a slower behavioral sequence, for example movement of an animal through a sequence of place fields, into a faster time scales associated with synaptic plasticity. The authors examine the synaptic weight change between pairs of neurons encoding different events in the behavioral sequence and find that phase precession enhances sequence learning when the learning rule is asymmetric over a relatively narrow time window (assuming the behavioral events encoded by the two neurons overlap, ie the place fields of the neurons overlap). For wider time windows, however, phase precession does not appear to convey any advantage.I thought the study was interesting – the idea that phase precession "compresses" sequences into theta cycles has been around for a bit, but this is the first study that I've seen that does analysis at this level. I think many researchers who are interested in temporal coding would find the work very interesting.

We thank the reviewer for the very positive comments.

I did, however, have a little trouble understanding what conclusions the study draws about the brain (if we are supposed to draw any). The authors conclude that phase precession facilitates if the learning window is shorter than a theta cycle – that seems in line with published STDPs rules from slice studies. However, Figure 4 seems to imply that the authors have recovered a 1 second learning window from Bittner's data – are they suggesting that phase precession is not an asset for the learning in that study (or did I miss something)?

Correct. In Figure 4C we compare the signal-to-noise ratio (SNR) of the weight change as a function of the width of the learning window. For asymmetric windows (colored solid lines), phase precession is helpful for temporal-order learning, but only for narrow (compared, e.g., to the theta oscillation cycle) learning windows. Interestingly, the SNR is increasing and saturates for larger widths of the learning window, which seems to suggest that very wide learning windows are optimal for temporal-order learning. To indicate that this behavior critically depends on the symmetry of the learning window, we included in this graph the SNR obtained with the learning window from Bittner’s data (dot marked “Bittner et al., (2017)” in the updated version of Figure 4), which has a strong symmetric component. In this case, the SNR is much lower than the SNR of an asymmetric window of the same width. Even though the Bittner window has a low SNR for temporal-order learning, it may still be useful for other tasks, for example place-field formation. The second-long learning rule seems to serve this purpose well. For temporal-order memory, however, it is not suited due to its strong symmetric component.

To indicate that there are published STDP rules that are more useful for temporal-order learning, we now include in the same graph (new Figure 4C) the SNR for several other experimentally observed learning windows. These windows have strong asymmetric components and - depending on the proportion of symmetric and asymmetric parts - can reach SNR values close to the theoretical prediction for perfectly asymmetric windows. This graph (and many other results presented in our study) suggests that phase precession in combination with experimentally determined narrow asymmetric learning windows could be a mechanism supporting temporal-order learning in hippocampal networks. This conclusion is also summarized in a similar way in the abstract.

Are there predictions to be made about how place fields must be spaced for optimal sequence learning?

Yes, for example Figure 3 predicts that for temporal-order learning with narrow learning windows there is an optimal overlap between firing fields. For asymmetric STDP windows, the maximum weight change (Figure 3A) and the maximum SNR (Figure 3C) are achieved for partially overlapping firing fields where the overlap *Tij* is in the range of the width σ of the firing fields. On the other hand, wide STDP windows support temporal-order learning only if they are largely asymmetric (Figure 4 and Figure 5) but in this case phase precession is not beneficial.

Also, I'd be curious to know how the authors analysis fits in with replay – is the assumption that neuromodulation is changing the time window or other learning dynamics?

A key assumption underlying our work is that neuromodulation affects the plasticity and the strength of synapses (see, e.g., the sections “Model assumptions” and “Width, shape, and symmetry of the STDP window…” in the Discussion). For example, acetylcholine (among other neuromodulators) seems to play a particular role by differentially modulating the distinct phases of memory encoding and memory consolidation. In our work we follow the idea [proposed by Hasselmo, (1999) and supported by many later studies] that during encoding excitatory feedback connections (and replay) are suppressed to avoid interference from previously stored information, but that the same synapses in this phase are highly plastic in order to store sequences; this may be mediated by high levels of acetylcholine. On the other hand, during slow-wave sleep, when acetylcholine levels are low and synapses are strong but less plastic, a sequence imprinted in recurrent synaptic weights can be replayed without having a too strong impact on the change of recurrent synaptic weights.

We have mentioned all these ideas on differential modulation of synapses and replay at various points in the manuscript. To better outline and summarize these important points in our manuscript, we have thoroughly revised and extended the Discussion.